# Unlocking the Power of Gradient Guidance for Structure-Based Molecule Optimization

## Abstract

Structure-based molecule optimization (SBMO) aims to optimize molecules with both continuous coordinates and discrete types against protein targets. A promising direction is to exert gradient guidance on generative models given its remarkable success in images, but it is challenging to guide discrete data and risks inconsistencies between modalities. To this end, we leverage a continuous and differentiable space derived through Bayesian inference, presenting ***Mol**ecule **J**oint **O**ptimization* (MolJO), the gradient-based SBMO framework that facilitates joint guidance signals across different modalities while preserving SE(3)-equivariance. We introduce a novel backward correction strategy that optimizes within a sliding window of the past histories, allowing for a seamless trade-off between explore-and-exploit during optimization. Our proposed MolJO achieves state-of-the-art performance on CrossDocked2020 benchmark (Success Rate 51.3%, Vina Dock -9.05 and SA 0.78), more than $4\times$ improvement in Success Rate compared to the gradient-based counterpart, and $2\times$ "Me-Better" Ratio as much as 3D baselines. Furthermore, we extend MolJO to a wide range of optimization settings, including multi-objective optimization and challenging tasks in drug design such as R-group optimization and scaffold hopping, further underscoring its versatility and potential.

## 1 Introduction

Structure-based drug design (SBDD) plays a critical role in drug discovery by identifying three-dimensional (3D) molecules that are favorable against protein targets (Isert et al., 2023). While recent SBDD focuses on the initial identification of potential drug candidates, these compounds must undergo a series of further modifications for optimized properties, a process that is both complex and time-consuming (Hughes et al., 2011). Therefore, structure-based molecule optimization (SBMO) has garnered increasing interest in real-world drug design (Zhou et al., 2024), emphasizing the practical need of optimizing 3D molecules to meet specific therapeutic criteria.

Concretely, SBMO can be viewed as a more advanced task within the broader scope of general SBDD, requiring precise control over molecular properties while navigating the chemical space. Specifically, SBMO addresses two key aspects: (1) *SBMO prioritizes targeted molecular property enhancement according to expert-specified objectives*, whereas generative models for SBDD primarily focus on maximizing data likelihood without special emphasis on property improvement (Luo et al., 2021; Peng et al., 2022). Therefore these models can only produce outputs similar to their training data, limiting the ability to improve molecular properties. (2) *SBMO is capable of optimizing existing compounds with 3D structural awareness*, addressing a critical gap left by previous molecule optimization methods with 1D SMILES or 2D graph representations (Bilodeau et al., 2022; Fu et al., 2022), and allowing for a more nuanced control. The focus on structure makes SBMO particularly suited for key design tasks, such as R-group optimization and scaffold hopping.

A pioneering work for SBMO is DecompOpt (Zhou et al., 2024), which designs a special 3D generative model conditioned on fragments, and employs a gradient-free sampling method within such decomposed fragment space through iterative oracle calls. One of the drawbacks is that it relies on oracle functions that can be computationally expensive especially in large-scale SBMO tasks, where running multiple rounds of costly simulations is impractical or even infeasible. Moreover, DecompOpt's contribution lies in its novel generative paradigm based on fragment conditions, rather than exploring optimization within existing models, making it less adaptable to other frameworks.

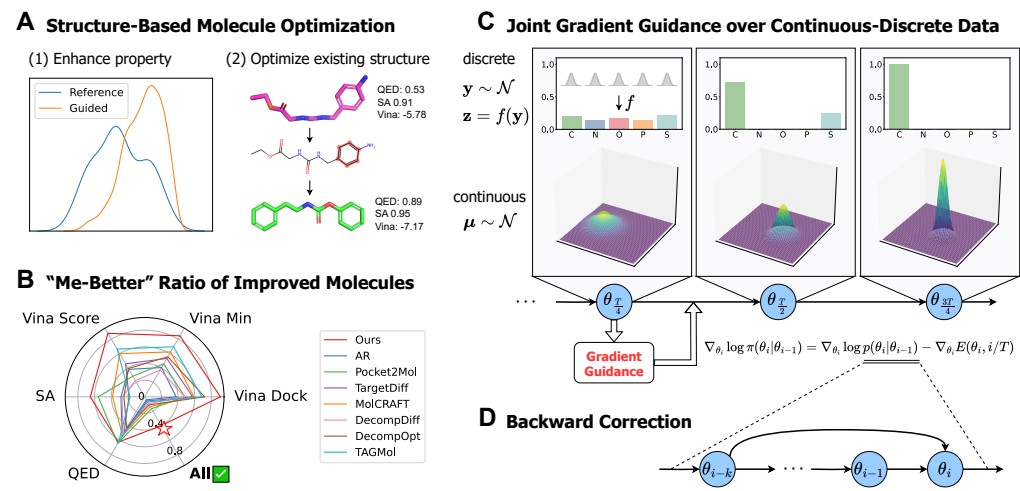

Figure 1: Overview. **A.** Structure-based molecule optimization, including (**1**) guiding molecule design by expert-specified objectives, (**2**) optimizing existing compounds in the structure space. **B.** Study on the ratio of "me-better" molecules, where all other baselines fall short in the overall improvement. **C.** Overall illustration of MolJO, utilizing joint gradient signals over continuous-discrete data, where the distributions of $\theta$ (continuous $\mu$ and discrete $z$) are taken from true guided trajectories. **D.** Graphical model of our proposed backward correction strategy, keeping a sliding window of size $k$.

One generalized solution is gradient guidance, addressing these issues by eliminating the need for expensive oracle simulations while being flexible enough to be incorporated into strong-performing generative models in a plug-and-play fashion, as demonstrated in a wide range of challenging real-world applications including image synthesis (Dhariwal & Nichol, 2021; Epstein et al., 2023).

However, current gradient-based methods have not fully realized their potential in SBMO, for they have historically suffered from the continuous-discrete challenges: (1) *it is non-trivial to guide discrete variable within probabilistic generative process.* More specifically, standard gradient guidance is designed for continuous variables that follow Gaussian distributions, making them not directly applicable to molecular data that involve discrete atom types. Methods attempting to adapt gradient guidance to discrete data often resort to approximating these variables as continuous, either by adding Gaussian noise (Bao et al., 2022) or by assuming that classifiers follow a Gaussian distribution (Vignac et al., 2023). Unfortunately, these approximations can lead to suboptimal results, as they do not accurately reflect the discrete nature (Kong et al., 2023). (2) *Gradient guidance might introduce inconsistencies between modalities.* For instance, TAGMol (Dorna et al., 2024) formalizes guidance exclusively over continuous coordinates, resulting in a disconnect between the discrete and continuous modalities. This may explain why TAGMol struggles to optimize overall molecular properties as shown in Figure 1, despite its improvement in Vina affinities. By solely guiding the continuous coordinates, TAGMol enhances spatial protein-ligand interactions but fails to optimize *e.g.* synthesizeability, which depends more on molecular topology, especially discrete atom types.

In this paper, we address the *multi-modality challenge for gradient guidance* by leveraging a continuous and differential space, representing an aggregation of noisy samples from the data space derived through Bayesian inference (Graves et al., 2023). We design MolJO (***M**olecule **J**oint **O**ptimization*), a general framework that enables gradient-based optimization of continuous and discrete variables in a structured manner. We introduce a novel sampling strategy called backward correction, enhancing the alignment of gradients over different steps. By maintaining a sliding window of past history for optimization, the backward correction strategy enforces explicit dependency on the past, effectively alleviating the issue of inconsistencies. Moreover, it balances the exploration of molecular space with the exploitation of better-aligned guidance signals, offering a flexible trade-off.

Our main contributions are summarized as follows:

- We propose MolJO, the joint gradient-based method for SBMO that establishes the guidance over molecules, offering better controllability and effectively integrating gradient guidance for continuous-discrete variables within a unified framework.

- We design a novel backward correction strategy for effective optimization. By keeping a sliding window and correcting the past given the current optimized version, we achieve better-aligned gradients and facilitate a flexible trade-off between exploration and exploitation.

- MolJO achieves the best Vina Dock of -9.05, SA of 0.78 and Success Rate of 51.3%, and "Me-Better" Ratio of improved molecules that is **$2\times$ as much as other 3D baselines**. We generalize MolJO to various needs including R-group optimization and scaffold hopping, highlighting its versatility and potential in practical drug design.

## 2 RELATED WORK

**Pocket-Aware Molecule Generation.** Pocket-aware generative models aim to learn a conditional distribution over the protein-ligand complex data. Initial approaches adopt 1D SMILES or 2D graph representation (Bjerrum & Threlfall, 2017; Gómez-Bombarelli et al., 2018), and recent research has shifted its focus towards 3D molecule generation in order to better capture the interatomic interactions. Early atom-based autoregressive models (Luo et al., 2021; Peng et al., 2022; Liu et al., 2022) enforce an atom ordering to generate molecules atom-by-atom. Fragment-based methods (Powers et al., 2022; Zhang et al., 2023; Lin et al., 2023) alleviate the issue of ordering by decomposing molecules into motifs instead of atom-level generation, but they risk more severe error accumulation and thus generally require post-processing or multi-stage treatment. Non-autoregressive methods based on diffusions (Schneuing et al., 2022; Guan et al., 2022; 2023) and BFNs (Qu et al., 2024) target full-atom generation for enhanced performance and efficiency. However, the needs of optimizing certain properties and modifying existing compounds are not adequately addressed in the scope of previous methods, limiting their usefulness in drug design.

**Gradient-Based Molecule Optimization.** Inspired by classifier guidance for diffusions (Dhariwal & Nichol, 2021), pioneering approaches are committed to adapting the guidance method to handle the complicated molecular geometries in the setting of pocket-unaware generation. EEGSDE (Bao et al., 2022) derives an equivariant framework for continuous diffusion, and MUDM (Han et al., 2023) further explores time-independent property functions for guidance. As they enforce a continuous diffusion process for discrete variables, these methods are not applicable in advanced molecular modeling (Guan et al., 2022; 2023) where discrete data are processed by a discrete diffusion, for it is unnatural to apply progressive Gaussian noise that drives Categorical data away from the simplex. DiGress (Vignac et al., 2023) proposes classifier guidance for discrete diffusion of molecular graphs, yet it additionally assumes that the probability of classifier follows a Gaussian, which is ungrounded and often a problematic approximation. Based on the continuous-discrete diffusion for SBDD, TAGMol (Dorna et al., 2024) retains the guidance only for continuous coordinates, because there lacks a proper way to propagate the gradient over discrete types. The discrete part is affected only implicitly and belatedly in the generative process, and such imbalanced guidance would probably result in suboptimal performance for lack of joint optimization.

## 3 PRELIMINARY

**3D Protein-Ligand Representation.** A protein binding site $\mathbf{p} = (\mathbf{x}_P, \mathbf{v}_P)$ is represented as a point cloud of $N_P$ atoms where atom coordinates $\mathbf{x}_P = \{\mathbf{x}_P^1, \ldots, \mathbf{x}_P^{N_P}\} \in \mathbb{R}^{N_P \times 3}$ and $K_P$-dimensional atom features $\mathbf{v}_P = \{\mathbf{v}_P^1, \ldots, \mathbf{v}_P^{N_P}\} \in \mathbb{R}^{N_P K_P}$. Similarly, a ligand molecule $\mathbf{m} = (\mathbf{x}_M, \mathbf{v}_M)$ contains $N_M$ atoms, where $\mathbf{x}_M^{(i)} \in \mathbb{R}^3$ is the atomic coordinate and $\mathbf{v}_M^{(i)} \in \mathbb{R}^{K_M}$ the atom type. For brevity, the subscript for molecules $\cdot_M$ and the pocket condition $\mathbf{p}$ are omitted unless necessary.

**Bayesian Flow Networks (BFNs).** Operating in a non-autoregressive fashion, BFN views the generative modeling as message exchange between a sender and a receiver. The *sender distribution* $p_S(\mathbf{y} \mid \mathbf{x}; \alpha)$ builds upon the accuracy level $\alpha$ applied to data $\mathbf{x}$ and defines the noised $\mathbf{y}$. The varying noise levels constitute the schedule $\beta(t) = \int_{t'=0}^{t} \alpha(t')dt'$, similar to that in diffusion models.

A key motivation for BFN is that the transmission ought to be continuous and smooth, therefore it does not directly operate on the noisy latent $\mathbf{y}$ as diffusions, but on the structured Bayesian posterior $\boldsymbol{\theta}$ given noisy latents instead. The receiver holds a prior belief $\boldsymbol{\theta}_0$, and updates the belief by aggregating

noisy observations $\mathbf{y}$, yielding the *Bayesian update distribution*:

$$p_U(\boldsymbol{\theta}_i \mid \boldsymbol{\theta}_{i-1}, \mathbf{x}; \alpha_i) = \mathop{\mathbb{E}}_{p_S(\mathbf{y}_i \mid \mathbf{x}; \alpha_i)} \delta\Big(\boldsymbol{\theta}_i - h(\boldsymbol{\theta}_{i-1}, \mathbf{y}_i, \alpha_i)\Big) \tag{1}$$

where $\delta(\cdot)$ is Dirac distribution, and Bayesian update function $h$ is derived through Bayesian inference. Note that $\boldsymbol{\theta}_i$ can be viewed as the result of mapping given the sender distributions of $\mathbf{y}_0$ to $\mathbf{y}_i$:

$$\boldsymbol{\theta}_i = f(\mathbf{y}_0, \mathbf{y}_1, \ldots, \mathbf{y}_i) \tag{2}$$

Intuitively, BFN aims to predict the clean sample given aggregated $\boldsymbol{\theta}$, *i.e.* conditioning on all previous latents. $\boldsymbol{\theta}$ is fed into a neural network $\boldsymbol{\Phi}$ to estimate the distribution of clean datapoint $\hat{\mathbf{x}}$, *i.e.* the *output distribution* $p_O(\hat{\mathbf{x}} \mid \boldsymbol{\Phi}(\boldsymbol{\theta}, t))$. The *receiver distribution* is obtained by marginalizing out $\hat{\mathbf{x}}$:

$$p_R(\mathbf{y}_i \mid \boldsymbol{\theta}_{i-1}; t_i, \alpha_i) = \mathop{\mathbb{E}}_{p_O(\hat{\mathbf{x}} \mid \boldsymbol{\Phi}(\boldsymbol{\theta}_{i-1}, t_i))} p_S(\mathbf{y}_i \mid \hat{\mathbf{x}}; \alpha_i) \tag{3}$$

The training objective is to minimize the KL-divergence between sender and receiver distributions:

$$L^n(\mathbf{x}) = \mathop{\mathbb{E}}_{\prod_{i=1}^n p_U(\boldsymbol{\theta}_i \mid \boldsymbol{\theta}_{i-1}, \mathbf{x}; \alpha_i)} \sum_{i=1}^n D_{\text{KL}}(p_S(\mathbf{y}_i \mid \mathbf{x}; \alpha_i) \,\|\, p_R(\mathbf{y}_i \mid \boldsymbol{\theta}_{i-1}, t_i, \alpha_i)). \tag{4}$$

## 4 METHOD

In this section, we introduce MolJO that guides the distribution over $\boldsymbol{\theta}$, utilizing aggregated information from previous latents. Though different from guided diffusions that operate on noisy latent $\mathbf{y}$, this guidance aligns with our generative process conditioned on $\boldsymbol{\theta}$. By focusing on $\boldsymbol{\theta}$, we can effectively steer the clean samples towards desirable direction, ensuring a smooth gradient flow.

**Notation.** Following Kong et al. (2024) and denoting the guided distribution $\pi$ as product of experts (Hinton, 2002) modulated by energy function $E$ that predicts certain property, we have $\pi(\boldsymbol{\theta}_i \mid \boldsymbol{\theta}_{i-1}) \propto p_\phi(\boldsymbol{\theta}_i \mid \boldsymbol{\theta}_{i-1}) p_E(\boldsymbol{\theta}_i)$, where $\boldsymbol{\Phi}$ is the pretrained network for BFN, $p_E(\boldsymbol{\theta}_i) = \exp\left[-E(\boldsymbol{\theta}_i, t_i)\right]$ is the unnormalized Boltzmann distribution corresponding to the time-dependent energy function.

**Overview.** As illustrated in Figure 1, we introduce MolJO as follows: in Sec. 4.1, we propose the concept of gradient guidance over the multi-modality molecule space, derive the form of guided transition kernel $\pi(\boldsymbol{\theta}_i \mid \boldsymbol{\theta}_{i-1})$ via first-order Taylor expansion, and explain the underlying manipulations of distributions the guidance corresponds to. In Sec. 4.2, we present a generalized advanced sampling strategy termed backward correction for $p_\phi$, which allows for a flexible trade-off between explore-and-exploit by maintaining a sliding window of past histories. We empirically demonstrate our strategy helps optimize consistency across steps, ultimately improving the overall performance.

### 4.1 EQUIVARIANT GUIDANCE FOR MULTI-MODALITY MOLECULAR DATA

In this section, we derive the detailed guidance over $\boldsymbol{\theta}$ for molecule $\mathbf{m} = (\mathbf{x}, \mathbf{v})$ with $N$ atoms, where $\mathbf{x} \in \mathbb{R}^{N \times 3}$ represent continuous atom coordinates and $\mathbf{v} \in \{1, \ldots, K\}^N$ for $K$ discrete atom types, and thus $\boldsymbol{\theta} := [\boldsymbol{\mu}, \mathbf{z}]$ for the continuous and discrete modality, respectively.

**Guidance over Multi-Modalities.** To steer the sampling process towards near-optimal samples, we utilize the score $\nabla_{\boldsymbol{\theta}} \log p_E(\boldsymbol{\theta})$ as a gradient-based property guidance, for which we have the following proposition (proof in Appendix C.1), followed by details for each modality.

**Proposition 4.1.** *Suppose $\boldsymbol{\mu}_i \sim \mathcal{N}(\boldsymbol{\mu}_\phi, \sigma\mathbf{I})$ and $\mathbf{y}_i \sim \mathcal{N}(\mathbf{y}_\phi, \sigma'\mathbf{I})$ in the original generative process of BFN, then we can approximate the guided transition kernel $\pi(\boldsymbol{\theta}_i \mid \boldsymbol{\theta}_{i-1})$ by sampling*

$$\boldsymbol{\mu}_i^* \sim \mathcal{N}(\boldsymbol{\mu}_\phi + \sigma\mathbf{g}_{\boldsymbol{\mu}}, \sigma\mathbf{I}) \tag{5}$$

$$\mathbf{y}_i^* \sim \mathcal{N}(\mathbf{y}_\phi + \sigma'\mathbf{g}_{\mathbf{y}}, \sigma'\mathbf{I}) \tag{6}$$

*where gradient $\mathbf{g}_{\boldsymbol{\mu}} = -\nabla_{\boldsymbol{\mu}} E(\boldsymbol{\theta}, t_i)|_{\boldsymbol{\theta} = \boldsymbol{\theta}_{i-1}}$, $\mathbf{g}_{\mathbf{y}} = -\nabla_{\mathbf{y}} E(\boldsymbol{\theta}, t_i)|_{\boldsymbol{\theta} = \boldsymbol{\theta}_{i-1}}$, recall $\boldsymbol{\theta} := [\boldsymbol{\mu}, \mathbf{z} = f(\mathbf{y})]$.*

The guidance is formalized over both continuous coordinates and discrete types, and differs from previous guided diffusion for molecules in that (1) it guides the discrete data through Gaussian-distributed latent $\mathbf{y}$ and ensures that the discrete variables are still on the probability simplex without relying on assumptions (Vignac et al., 2023) or relaxations (Bao et al., 2022; Han et al., 2023), and (2) alleviates the inconsistencies between modalities (Dorna et al., 2024) by joint gradient signals.

**Guiding $\boldsymbol{\mu}$ for Continuous x.** For continuous coordinates $\mathbf{x} \in \mathbb{R}^{N \times 3}$, it is natural to adopt a Gaussian sender distribution $\mathbf{y}^x \sim \mathcal{N}(\mathbf{x}, \alpha^{-1}\mathbf{I})$. With a prior belief $\boldsymbol{\mu}_0 = \mathbf{0}$, we have the Bayesian update function for posterior $\boldsymbol{\mu}_i$ given noisy $\mathbf{y}^x$ as in Graves et al. (2023):

$$h(\boldsymbol{\mu}_{i-1}, \mathbf{y}^x, \alpha_i) = \frac{\boldsymbol{\mu}_{i-1}\rho_{i-1} + \mathbf{y}^x \alpha_i}{\rho_i} \tag{7}$$

with $\alpha_i = \beta_i - \beta_{i-1}, \rho_i = 1 + \beta_i$ given the schedule $\beta_i = \sigma_1^{-2i/n} - 1$ for a positive $\sigma_1$ and $n$ steps.

*Remark* 4.2. In the continuous domain, guidance over $\boldsymbol{\theta}$ (*i.e.* $\boldsymbol{\mu}$) is analogous to guided diffusions, since guiding $\boldsymbol{\mu}$ corresponds to guiding noisy latent $\mathbf{y}$ using the gradient over it:

$$\boldsymbol{\mu}_i^* = \boldsymbol{\mu}_i + \sigma \mathbf{g}_{\boldsymbol{\mu}} = \frac{\boldsymbol{\mu}_{i-1}\rho_{i-1} + (\mathbf{y}^x + \sigma \frac{\rho_i}{\alpha_i}\mathbf{g}_{\boldsymbol{\mu}})\alpha_i}{\rho_i} = \frac{\boldsymbol{\mu}_{i-1}\rho_{i-1} + (\mathbf{y}^x + \sigma \mathbf{g}_{\mathbf{y}^x})\alpha_i}{\rho_i} \tag{8}$$

where by chain rule $\mathbf{g}_{\mathbf{y}^x} = -\nabla_{\mathbf{y}^x} E(\boldsymbol{\theta}, t_i)|_{\boldsymbol{\theta}=\boldsymbol{\theta}_{i-1}} = \mathbf{g}_{\boldsymbol{\mu}} \frac{\partial h}{\partial \mathbf{y}^x} = \mathbf{g}_{\boldsymbol{\mu}} \frac{\alpha_i}{\rho_i}$. This builds a connection between the guidance over $\boldsymbol{\theta}$ and over latent $\mathbf{y}$ for continuous data.

**Guiding z for Discrete v.** For $N$-dimensional discrete types $\mathbf{v} \in \{1, \ldots, K\}^N$, the noisy latent represents the counts of each type among $K$ types, where we have $\mathbf{y}^v \sim \mathcal{N}(\mathbf{y}^v | \alpha'(K\mathbf{e_v} - \mathbf{1}), \alpha' K\mathbf{I})$, $\mathbf{e_v} = [\mathbf{e}_{\mathbf{v}^{(1)}}, \ldots, \mathbf{e}_{\mathbf{v}^{(N)}}] \in \mathbb{R}^{KN}$, $\mathbf{e}_{\mathbf{v}^{(j)}} = \delta_{\mathbf{v}^{(j)}} \in \mathbb{R}^K$ with Kronecker delta function $\delta$. This is mathematically grounded with further explanation in Appendix B.

The aggregated $\mathbf{z}_i$ as a posterior belief is updated given the prior $\mathbf{z}_0 = \frac{\mathbf{1}}{\mathbf{K}}$ as:

$$h(\mathbf{z}_{i-1}, \mathbf{y}^v, \alpha_i') = \frac{\exp(\mathbf{y}^v)\mathbf{z}_{i-1}}{\sum_{k=1}^K \exp(\mathbf{y}_k^v)(\mathbf{z}_{i-1})_k} \tag{9}$$

where the redundant $\alpha_i' = \beta_i' - \beta_{i-1}'$ with $\beta_i' = \beta_1'(\frac{i}{n})^2$, given a positive hyperparameter $\beta_1'$.

*Remark* 4.3. In the discrete domain, guiding all latents $\mathbf{y}$ amounts to a reweight of the Categorical distribution for $\boldsymbol{\theta}$ (*i.e.* $\mathbf{z}$), changing the probability of each class in accordance with the gradient. Take an extreme case to illustrate, where $\mathbf{g}_{\mathbf{y}^v}$ is filled with one-hot vectors $\delta_d$:

$$(\mathbf{z}_i^*)_k = \frac{\exp(\mathbf{y}_k^v)(\mathbf{z}_{i-1})_k}{\sum_{k'=1}^K \exp(\mathbf{y}_{k'}^v)(\mathbf{z}_{i-1})_{k'} + [\exp(\sigma') - 1]\exp(\mathbf{y}_d^v)(\mathbf{z}_{i-1})_d} = (\mathbf{z}_i)_k \frac{1}{1+C} < (\mathbf{z}_i)_k \tag{10}$$

for all $k \neq d$, where $C = [\exp(\sigma') - 1]\exp(\mathbf{y}_d^v)(\mathbf{z}_{i-1})_d / [\sum_{k'=1}^K \exp(\mathbf{y}_{k'}^v)] > 0$ as the variance $\sigma' > 0$. It is obvious that the guidance lowers the probability for all classes but the favored $d$, redistributing the mass for discrete data in a more structured way than diffusion counterparts.

**Equivariance.** Our proposed guided sampling that utilizes joint gradient signals is still equivariant as shown in the proposition below, with proof in Appendix C.2.

**Proposition 4.4.** *The guided sampling process preserves SE(3)-equivariance when $\boldsymbol{\Phi}$ is SE(3)-equivariant, if the energy function $E(\boldsymbol{\theta}, \mathbf{p}, t)$ is also parameterized with an SE(3)-equivariant neural network, and the complex is shifted to the space where the protein's Center of Mass (CoM) is zero.*

### 4.2 GUIDED BAYESIAN UPDATE WITH BACKWARD CORRECTION

Here we propose a general backward correction sampling strategy inspired from the optimization perspective, and analyze its effect on aligning the gradients. Recall that from Eq. 1 we can aggregate $\boldsymbol{\theta}_i$ based on latents from the previous step:

$$p_\phi(\boldsymbol{\theta}_i \mid \boldsymbol{\theta}_{i-1}) = \mathbb{E}_{p_O(\hat{\mathbf{x}}_i | \boldsymbol{\Phi}(\boldsymbol{\theta}_{i-1}, t_i))} p_U(\boldsymbol{\theta}_i \mid \boldsymbol{\theta}_{i-1}, \hat{\mathbf{x}}_i; \alpha_i) \tag{11}$$

Backward correction aims at "*correcting the past to further optimize*". Since we obtain an optimized $\boldsymbol{\theta}_i^*$ from the guided kernel $\pi(\boldsymbol{\theta}_i | \boldsymbol{\theta}_{i-1})$, there will be an optimized version of $\hat{\mathbf{x}}_i^* = \hat{\mathbf{x}}_{i+1}$ for the next step. By backward correcting the Bayesian update distribution $p_U$ given the optimized $\hat{\mathbf{x}}^*$, we are able to reinforce the current best possible parameter $\boldsymbol{\theta}$, instead of building on the suboptimal history.

---

**Algorithm 1** Gradient Guided Sampling of MolJO with Backward Correction

---

**Require:** network $\boldsymbol{\Phi}(\boldsymbol{\theta}, t, \mathbf{p})$, noise schedules $[\beta(t), \beta'(t)]$, number of sample steps $n$, back correction steps $k$, number of atom types $K$, energy function $E(\boldsymbol{\theta}, \mathbf{p}, t)$, guidance scale $s$
1: Initialize belief $\boldsymbol{\theta} := [\boldsymbol{\mu}, \mathbf{z}] \leftarrow [\mathbf{0}, \frac{1}{K}]$     ▷ Continuous aggregations for different modalities
2: **for** $i = 1$ to $n$ **do**
3:   $[t, t_{-k}] \leftarrow [\frac{i-1}{n}, \max(0, \frac{i-k-1}{n})]$        ▷ Discrete timesteps in $[0, 1)$
4:   $[\hat{\mathbf{x}}, \hat{\mathbf{e}}_{\mathbf{v}}] \leftarrow \boldsymbol{\Phi}(\boldsymbol{\theta}, t, \mathbf{p})$          ▷ Network predicts clean data
5:   $[\mathbf{g}_{\boldsymbol{\mu}}, \mathbf{g}_{\mathbf{y}}] \leftarrow [-\nabla_{\boldsymbol{\mu}} E(\boldsymbol{\theta}, \mathbf{p}, t), -\nabla_{\mathbf{y}} E(\boldsymbol{\theta}, \mathbf{p}, t)]$     ▷ Gradient guidance
6:   $[\rho, \rho_{-k}, \Delta\beta, \Delta\beta'] \leftarrow [1 + \beta(t), 1 + \beta(t_{-k}), \beta(t) - \beta(t_{-k}), \beta'(t) - \beta'(t_{-k})]$
7:   Retrieve $\boldsymbol{\mu}_{-k}, \mathbf{z}_{-k}$ from the past      ▷ Backward correction starting point
8:   Sample $\boldsymbol{\mu} \sim \mathcal{N}([\frac{\rho(\Delta\beta\hat{\mathbf{x}} + \boldsymbol{\mu}_{-k}\rho_{-k}) + s\Delta\beta\mathbf{g}_{\boldsymbol{\mu}}}{\rho^2}, \frac{\Delta\beta}{\rho^2}\mathbf{I})$    ▷ Guidance for continuous $\mathbf{x}$
9:   Sample $\mathbf{y} \sim \mathcal{N}(\Delta\beta'(t)(K\hat{\mathbf{e}}_{\mathbf{v}} - \mathbf{1} + sK\mathbf{g}_{\mathbf{y}}), \Delta\beta'(t)K\mathbf{I})$    ▷ Guidance for discrete $\mathbf{v}$
10:   Update $\mathbf{z} \leftarrow \frac{\exp(\mathbf{y})\mathbf{z}_{-k}}{\sum_{i=1}^{K} \exp(\mathbf{y}_i)(\mathbf{z}_{-k})_i}$      ▷ Aggregate latents for discrete data
11: **end for**
12: $[\hat{\mathbf{x}}, \hat{\mathbf{e}}_{\mathbf{v}}] \leftarrow \boldsymbol{\Phi}(\boldsymbol{\theta}, 1, \mathbf{p})$         ▷ Network predicts final output
13: $\hat{\mathbf{v}} \leftarrow \arg\max(\hat{\mathbf{e}}_{\mathbf{v}})$         ▷ Final discrete output from p.m.f.
14: **return** $[\hat{\mathbf{x}}, \hat{\mathbf{v}}]$

---

By utilizing the property of additive accuracy once $p_U$ follows certain form as described by Graves et al. (2023), the one-step backward correction can be derived as follows:

$$p_\phi(\boldsymbol{\theta}_i \mid \boldsymbol{\theta}_{i-1}, \boldsymbol{\theta}_{i-2}) = \underset{p_O(\hat{\mathbf{x}}_i|\boldsymbol{\Phi}(\boldsymbol{\theta}_{i-1}, t_i))}{\mathbb{E}} \underset{p_U(\boldsymbol{\theta}_{i-1}|\boldsymbol{\theta}_{i-2}, \hat{\mathbf{x}}_i; \alpha_{i-1})}{\mathbb{E}} p_U(\boldsymbol{\theta}_i \mid \boldsymbol{\theta}_{i-1}, \hat{\mathbf{x}}_i; \alpha_i)$$

$$\text{originally } \hat{\mathbf{x}}_{i-1} \sim p_O(\hat{\mathbf{x}}_{i-1}|\boldsymbol{\Phi}(\boldsymbol{\theta}_{i-2}, t_{i-1}))$$

$$= \underset{p_O(\hat{\mathbf{x}}_i|\boldsymbol{\Phi}(\boldsymbol{\theta}_{i-1}, t_i))}{\mathbb{E}} p_U(\boldsymbol{\theta}_i \mid \boldsymbol{\theta}_{i-2}, \hat{\mathbf{x}}_i; \alpha_{i-1} + \alpha_i) \tag{12}$$

and we arrive at the $k - 1$ step corrected estimation of $p_\phi$:

$$p_\phi(\boldsymbol{\theta}_n|\boldsymbol{\theta}_{n-1}, \boldsymbol{\theta}_{n-k}) = \underset{p_O(\hat{\mathbf{x}}_n|\boldsymbol{\Phi}(\boldsymbol{\theta}_{n-1}, t_n))}{\mathbb{E}} p_U(\boldsymbol{\theta}_n|\boldsymbol{\theta}_{n-k}, \hat{\mathbf{x}}_n; \sum_{i=n-k+1}^{n} \alpha_i) \tag{13}$$

Plugging Eq. 7 and 9 together with the sender distributions defined above into the right hand side according to Eq. 1, yields the form of the backward corrected Bayesian update

$$p_U(\boldsymbol{\mu}_n|\boldsymbol{\mu}_{n-k}, \hat{\mathbf{x}}_n) = \mathcal{N}\left(\frac{\Delta\beta\hat{\mathbf{x}}_n + \boldsymbol{\mu}_{n-k}\rho_{n-k}}{\rho_n}, \frac{\Delta\beta}{\rho_n^2}\mathbf{I}\right) \tag{14}$$

$$p_U(\mathbf{z}_n|\mathbf{z}_{n-k}, \hat{\mathbf{v}}_n) = \underset{\mathbf{y} \sim \mathcal{N}(\mathbf{y}|\Delta\beta'(K\mathbf{e}_{\hat{\mathbf{v}}_n} - \mathbf{1}), \Delta\beta'K\mathbf{I})}{\mathbb{E}} \delta\left(\mathbf{z}_n - \frac{\exp(\mathbf{y})\mathbf{z}_{n-k}}{\sum_{i=1}^{K} \exp(\mathbf{y}_i)(\mathbf{z}_{n-k})_i}\right) \tag{15}$$

where $\hat{\mathbf{m}} = [\hat{\mathbf{x}}, \hat{\mathbf{v}}]$ is drawn from the output distribution $p_O(\hat{\mathbf{m}} \mid \boldsymbol{\Phi}(\boldsymbol{\theta}_{i-1}, t_{i-1}, \mathbf{p}))$ given pocket $\mathbf{p}$, $\Delta\beta = \beta_n - \beta_{n-k}$ and $\Delta\beta' = \beta'_n - \beta'_{n-k}$ are obtained from corresponding schedules.

The concept of sliding window unifies different sampling strategies proposed by Graves et al. (2023) (k=1) and Qu et al. (2024) (k=n). To understand its effect, we visualize the cosine similarity of gradients at each step w.r.t. the previous step in Figure 2. By changing the size $k$ of sliding window, it succeeds in balancing sample quality (*explore*) and optimization efficiency (*exploit*), where it first focuses on exploring the molecular space with rapidly changing structures and gradients, and then exploits better-aligned guidance signals over gradually refined structures, achieving the best success rate as later shown.

In practice, we employ the gradient scale $s$ as a temperature parameter, which is equivalent to adopting $p_E^s(\boldsymbol{\theta}, t) \propto \exp[-sE(\boldsymbol{\theta}, t)]$. The general sampling procedure is summarized in Algorithm 1.

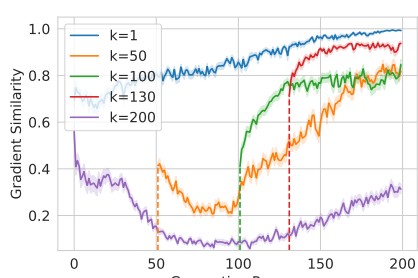

Figure 2: Gradient similarity, where $k$ denotes backward correction window size. For $1 < k < 200$, the similarity before timestep $k$ is omitted for it overlaps with $k = 200$, *i.e.* covering all the past.

## 5 EXPERIMENTS

### 5.1 EXPERIMENTAL SETUP

We conduct two sets of experiments for structure-based molecule optimization (SBMO), although the constrained setting seems within the scope of unconstrained one, it is biologically meaningful and more practical in rational drug design, and further showcases the flexibility of our method.

**Task.** For a molecule $\mathbf{m} \in \mathcal{M}$ where $\mathcal{M}$ denotes the set of molecules, there are oracles $a_i(\mathbf{m})$ : $\mathcal{M} \to \mathbb{R}$ for property $i$, each with a desired threshold $\delta_i \in \mathbb{R}$. MolJO is capable of different levels of controllability: (1) unconstrained optimization, where we identify a set of molecules such that $\{\mathbf{m} \in \mathcal{M} \mid a_i(\mathbf{m}) \geq \delta_i, \forall i\}$, *i.e.* the goal is to optimize a number of objectives. (2) constrained optimization, where we aim to find a set of molecules that contain specific substructures $s$ such that $\{\mathbf{m} \in \mathcal{M} \mid a_i(\mathbf{m}) \geq \delta_i, s \subset \mathbf{m}, \forall i\}$.

**Dataset.** Following previous SBDD works (Luo et al., 2021; Peng et al., 2022; Guan et al., 2022), we utilize CrossDocked2020 (Francoeur et al., 2020) to train and test our model, and adopt the same processing that filters out poses with RMSD $> 1$Å and clusters proteins based on 30% sequence identity, yielding 100,000 training poses and 100 test proteins.

**Baselines.** We divide all baselines into the following: (1) Generative models (*Gen.*), including AR (Luo et al., 2021), GraphBP (Liu et al., 2022), Pocket2Mol (Peng et al., 2022), FLAG (Zhang et al., 2023), DiffSBDD (Schneuing et al., 2022), TargetDiff (Guan et al., 2022), DecompDiff (Guan et al., 2023), IPDiff (Huang et al., 2024) and MolCRAFT (Qu et al., 2024), (2) Oracle-based optimization (*Oracle Opt.*) that rely on docking simulation in each round, such as AutoGrow4 (Spiegel & Durrant, 2020), RGA (Fu et al., 2022), and DecompOpt (Zhou et al., 2024), (3) Gradient-guided (*Grad. Opt.*) TAGMol (Dorna et al., 2024). Detailed descriptions of baselines are left in Appendix F.

**Metrics.** We employ the commonly used metrics as follows: (1) Affinity metrics calculated by Autodock Vina (Eberhardt et al., 2021), in which **Vina Score** calculates the raw energy of the given molecular pose residing in the pocket, **Vina Min** conducts a quick local energy minimization and scores the minimized pose, and **Vina Dock** performs a relatively longer search for optimal pose to calculate the lowest energy. **Success Rate** measures the percentage of generated molecules that pass certain criteria (Vina Dock $< -8.18$, QED $> 0.25$, SA $> 0.59$) following Long et al. (2022) and Guan et al. (2022). (2) Molecular properties, including drug-likeness (**QED**) and synthesizability score (**SA**). (3) Metrics for sample distribution, such as diversity (**Div**). A more comprehensive set of metrics are detailed in Appendix F.

### 5.2 UNCONSTRAINED OPTIMIZATION

In this section, we demonstrate the ability of our framework to improve molecular properties in both single and multi-objective optimization. We sample 100 molecules for each protein and evaluate MolJO in optimizing binding affinity and molecular properties. For additional evaluation of molecular conformation besides optimization performance, please see Appendix G.

**MolJO effectively enhances molecular property w.r.t. generative models.** The optimized distribution greatly improves upon the original generated distribution, as shown in the distribution shift in Figure 3 for single objective optimization, and Table 1 (row 14 *vs.* row 9).

**MolJO outperforms gradient-based method with $4\times$ higher Success Rate.** As shown in Table 1, our model achieves state-of-the-art in affinity-related metrics while being highly drug-like, with the best Success Rate of 51.3%, a fourfold improvement over TAGMol (row 14 *vs.* row 13).

**MolJO has more potential than oracle-based baselines if equipped with oracles.** RGA (Fu et al., 2022) and DecompOpt (Zhou et al., 2024) show satisfactory Success Rate, enjoying the advantage of oracle-based screening at some expense of diversity, while AutoGrow4 (Spiegel & Durrant, 2020) falls short in QED, yielding a suboptimal Success Rate. Given the same concentration use of Z-score (Zhou et al., 2024), we report a variant of MolJO with top-of-$N$, selecting a tenth

Table 1: Summary of different properties of reference molecules and generated molecules by our model and other baselines, where G+O denotes equipping our method with top-of-$N$ in oracle simulations. (↑) / (↓) denotes a larger / smaller number is better. Top 2 results are highlighted with **bold text** and underlined text.

| | Methods | Vina Score (↓) | | Vina Min (↓) | | Vina Dock (↓) | | QED (↑) | SA (↑) | Div (↑) | Success |
|---|---|---|---|---|---|---|---|---|---|---|---|
| | | Avg. | Med. | Avg. | Med. | Avg. | Med. | Avg. | Avg. | Avg. | Rate (↑) |
| | Reference | -6.36 | -6.46 | -6.71 | -6.49 | -7.45 | -7.26 | 0.48 | 0.73 | - | 25.0% |
| Gen. | ① AR | -5.75 | -5.64 | -6.18 | -5.88 | -6.75 | -6.62 | 0.51 | 0.63 | 0.70 | 6.9% |
| | ② GraphBP | - | - | - | - | -4.80 | -4.70 | 0.43 | 0.49 | **0.79** | 0.1% |
| | ③ Pocket2Mol | -5.14 | -4.70 | -6.42 | -5.82 | -7.15 | -6.79 | 0.57 | 0.76 | 0.69 | 24.4% |
| | ④ FLAG | 45.85 | 36.52 | 9.71 | -2.43 | -4.84 | -5.56 | 0.61 | 0.63 | 0.70 | 1.8% |
| | ⑤ DiffSBDD | -1.44 | -4.91 | -4.52 | -5.84 | -7.14 | -7.30 | 0.47 | 0.58 | 0.73 | 7.9% |
| | ⑥ TargetDiff | -5.47 | -6.30 | -6.64 | -6.83 | -7.80 | -7.91 | 0.48 | 0.58 | 0.72 | 10.5% |
| | ⑦ DecompDiff | -5.19 | -5.27 | -6.03 | -6.00 | -7.03 | -7.16 | 0.51 | 0.66 | 0.73 | 14.9% |
| | ⑧ IPDiff | -6.41 | -7.01 | -7.45 | -7.48 | -8.57 | -8.51 | 0.52 | 0.59 | 0.74 | 16.5% |
| | ⑨ MolCRAFT | -6.55 | -6.95 | -7.21 | -7.14 | -7.67 | -7.82 | 0.50 | 0.67 | 0.70 | 26.8% |
| Oracle Opt. | ⑩ AutoGrow4 | - | - | - | - | -8.99 | -9.00 | 0.46 | 0.76 | 0.47 | 14.3% |
| | ⑪ RGA | - | - | - | - | -8.01 | -8.17 | 0.57 | 0.71 | 0.41 | 46.2% |
| | ⑫ DecompOpt | -5.75 | -5.97 | -6.58 | -6.70 | -7.63 | -8.02 | 0.56 | 0.73 | 0.63 | 39.4% |
| Grad. Opt. | ⑬ TAGMol | -7.02 | -7.77 | -7.95 | -8.07 | -8.59 | -8.69 | 0.55 | 0.56 | 0.69 | 11.1% |
| | ⑭ MolJO | -7.52 | -8.02 | -8.33 | -8.34 | -9.05 | -9.13 | 0.56 | 0.78 | 0.66 | 51.3% |
| G + O | ⑮ MolJO† (N=10) | **-8.54** | **-8.81** | **-9.48** | **-9.09** | **-10.50** | **-10.14** | **0.67** | **0.79** | 0.61 | **70.3%** |

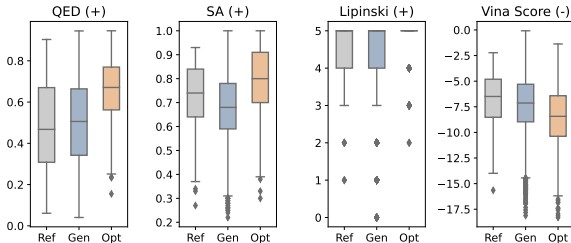

Figure 3: Distribution shift from test set (Ref), backbone without guidance (Gen) to guided MolJO (Opt).

Table 2: Properties of molecules with a larger average size, where Vina stands for Vina Dock Avg., SR for Success Rate. Top-1 results are highlighted with **bold text**.

| Methods | Vina | QED | SA | SR | Size |
|---|---|---|---|---|---|
| Reference | -7.45 | 0.48 | 0.73 | 25.0% | 22.8 |
| DecompDiff | -8.39 | 0.45 | 0.61 | 24.5% | 29.4 |
| DecompOpt | -9.01 | 0.48 | 0.65 | 52.5% | 32.9 |
| MolCRAFT | -9.25 | 0.46 | 0.62 | 36.6% | 29.4 |
| MolJO | **-10.53** | **0.50** | **0.72** | **64.2%** | 30.0 |

of top scoring molecules and showing that it is more effective than oracle-based methods once in a similar setting. Moreover, the higher diversity of DecompOpt and MolJO suggests the superiority of 3D structure-aware generative models over 2D optimization baselines (row 15 *vs.* row 10-12).

**MolJO is $2\times$ as effective in proposing "me-better" candidates.** For gradient-based method TAGMol (Dorna et al., 2024), although it produces seemingly promising high affinity binders, they come at the expense of sacrificed molecular properties like QED and SA, demonstrating the suboptimal control of coordinate-only guidance signals. Notably, the ratio of all-better samples is below 17% for all other baselines, and MolJO is twice as effective (39.8%) in generating feasible drug candidates that pass this criteria, as shown in Figure 1.

**MolJO excels even in optimizing large OOD molecules.** Note that for fair comparison, we restrict the size of generated molecules by reference molecules so that both generative models and optimization methods navigate the similar chemical space, as we observe a clear correlation between properties and sizes in Figure 6. For model variants capable of exploring larger number of atoms, we report the results in Table 2 with sizes, where MolJO consistently outperforms other baselines, demonstrating its robustness. A detailed discussion can be found in Appendix E.

## 5.3 CONSTRAINED OPTIMIZATION

Constrained optimization seeks to optimize the input reference molecules for enhanced properties while retaining specific structures. We generalize our framework with such structural control and

Table 3: Constrained optimization results, where *Redesign* means R-group optimization with fragments of the same size redesigned, *Growing* means fragment growing into larger size, *Hopping* means scaffold hopping. (↑) / (↓) indicates a larger / smaller number is better. Top-1 highlighted in **bold**.

| Methods | | Vina Score (↓) | | Vina Min (↓) | | Vina Dock (↓) | | QED (↑) | SA (↑) | Connected | Success |
|---|---|---|---|---|---|---|---|---|---|---|---|
| | | Avg. | Med. | Avg. | Med. | Avg. | Med. | Avg. | Avg. | Avg. (↑) | Rate (↑) |
| Reference | | -6.36 | -6.46 | -6.71 | -6.49 | -7.45 | -7.26 | 0.48 | 0.73 | 100% | 25.0% |
| Redesign | TargetDiff | -6.14 | -6.21 | -6.79 | -6.58 | -7.70 | -7.61 | 0.50 | 0.64 | 85.5% | 18.9% |
| | TAGMol | -6.60 | -6.66 | -7.10 | -6.80 | -7.63 | -7.76 | 0.53 | 0.62 | 87.0% | 19.2% |
| | MolCRAFT | -6.63 | -6.70 | -7.12 | -6.91 | -7.79 | -7.72 | 0.49 | 0.67 | **96.7%** | 22.7% |
| | MolJO | **-7.13** | **-7.28** | **-7.62** | **-7.39** | **-8.16** | **-8.20** | **0.57** | **0.68** | 95.1% | **29.0%** |
| Growing | TargetDiff | -6.73 | -7.29 | -7.60 | -7.67 | -8.89 | -8.79 | 0.39 | 0.52 | 71.6% | 11.2% |
| | TAGMol | -7.30 | -7.70 | -8.08 | -7.81 | -8.92 | -8.78 | 0.47 | 0.53 | 78.7% | 11.8% |
| | MolCRAFT | -6.96 | -7.47 | -7.86 | -7.73 | -8.80 | -8.65 | 0.44 | 0.59 | 91.7% | 19.9% |
| | MolJO | **-8.08** | **-8.35** | **-8.79** | **-8.58** | **-9.21** | **-9.45** | **0.53** | **0.62** | **93.2%** | **32.7%** |
| Hopping | TargetDiff | -5.72 | -5.78 | -6.00 | -5.83 | -6.31 | -6.66 | 0.39 | 0.65 | 63.3% | 6.2% |
| | TAGMol | -6.17 | -6.10 | -6.46 | -6.07 | -7.19 | -6.80 | 0.44 | 0.62 | 68.7% | 6.9% |
| | MolCRAFT | -6.31 | -6.17 | -6.58 | -6.40 | -7.25 | -7.15 | 0.42 | 0.67 | 89.9% | 14.6% |
| | MolJO | **-6.86** | **-6.50** | **-7.13** | **-6.70** | **-7.67** | **-7.58** | **0.46** | **0.68** | **90.5%** | **23.6%** |

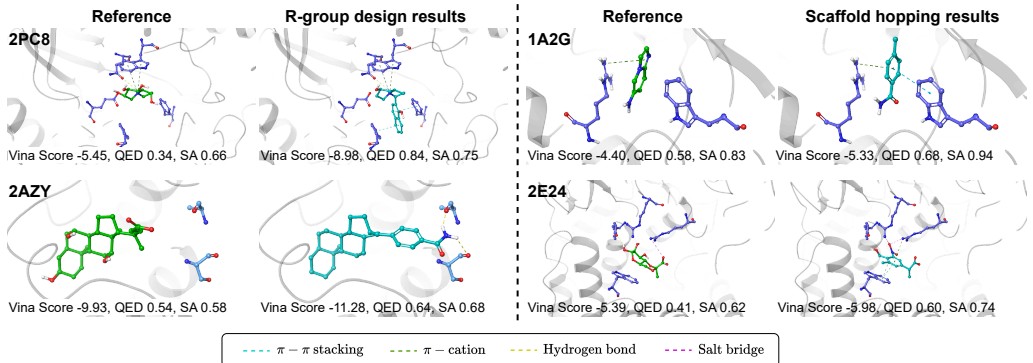

Figure 4: Visualization of the binding modes of the reference molecule (carbons in green) and the optimized molecule (in cyan) within the protein pocket (PDB ID: 2PC8, 2AZY, 1A2G, 2E24). The molecules and key residues (in blue) are shown in stick, while the protein's main chain is drawn in cartoon (in gray). Dashed lines of various colors indicate different types of non-bonding interactions. **Left:** R-group optimization results. **Right:** scaffold hopping results.

show its potential for pharmaceutical use cases including R-group optimization and scaffold hopping, achieved by infilling. Details of this task are in Appendix D.2.

**MolJO captures the complex environment of infilling.** Table 3 shows that our method generates valid connected molecules and captures the complicated chemical environment with better molecular properties than all baselines, showcasing its potential for lead optimization. As for diffusion baselines, they generate fewer valid connected molecules especially in the challenging case with scaffold hopping, with diffusion baselines lower than 70% validity, and proves to be less effective in proposing feasible candidates, with Success Rate < 20%.

**Optimized molecules form more key interactions for binding.** The visualization for constrained optimization is shown in Figure 4. It can be seen that the optimized molecules establish more key interactions with the protein pockets, thus binding more tightly to the active sites. For example, the optimized molecule for 2PC8 retains the key interaction formed by its scaffold, with R-group grown deeper inside the pocket, forming another two π-π stackings.

## 5.4 ABLATION STUDIES

We conduct ablation studies to thoroughly validate our design. More details are left in Appendix F.2.

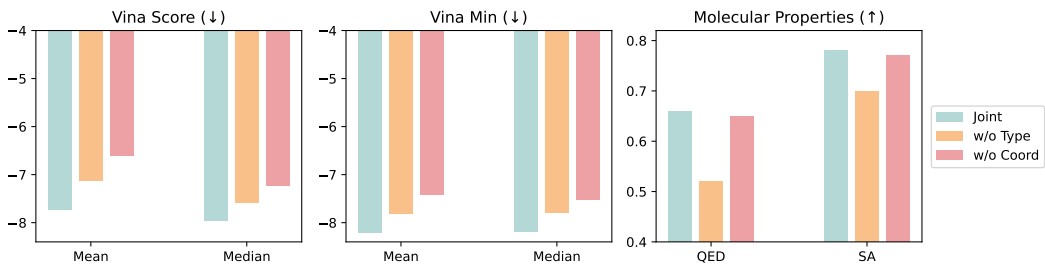

Figure 5: Ablation studies of joint optimization over atom types and coordinates, where *w/o type* means the gradient is disabled for types, *w/o coord* disables the gradient for coordinates.

**Joint guidance is consistently better than single-modality guidance.** To validate our choice of joint guidance over different modalities, we ablate the gradient for coordinates or types. As shown in Figure 5, utilizing gradients to guide both data modalities is consistently better than applying single-modality gradient only. For affinities, optimizing coordinates is effective in improving the spatial interactions, while for drug-like properties, guidance over atom types plays a crucial role. This underscores the significance of deriving appropriate guidance form jointly, and again supports our finding that a single coordinate guidance as in TAGMol is insufficient and yields suboptimal results.

**Backward correction boosts both the unguided sampling and the effect of guidance.** We denote sampling $\boldsymbol{\theta}_i$ according to Eq. 11 *Vanilla* for sampling only one $y \sim p_S$ in Eq. 1, *Vanilla MC* for a Monte Carlo estimate that integrates over multiple $y$, and sampling proposed by MolCRAFT is denoted *Full B.C.* as it corresponds to setting $k = n = 200$ in Eq. 13, and for *B.C.* we set $k = 130$ as backward correction steps. Table 4 shows that our method of keeping a history and selectively correcting the past not only improves the original unguided sampling, but also yields better results with guidance. Note that for vanilla case, the gradient guidance does not work as much probably due to the suboptimal history, while correcting a sufficient number of past steps boosts the optimization.

Table 4: Performances of no backward correction (*Vanilla & Vanilla MC*), fully corrected strategy (*Full B.C.*) and backward correction strategy (*B.C.*). Positive numbers in green show the relative improvement under guidance, while non-positive numbers in black indicate no performance gain.

| | Sampling Strategy | Vina Score (↓) | | Vina Min (↓) | | QED (↑) | SA (↑) |
| --- | --- | --- | --- | --- | --- | --- | --- |
| | | Avg. | Med. | Avg. | Med. | Avg. | Avg. |
| w/o Guide | Vanilla (k=1) | -5.23 | -5.81 | -6.30 | -6.17 | 0.46 | 0.62 |
| | Vanilla MC (k=1) | -6.25 | -6.70 | -7.01 | -7.05 | **0.51** | 0.61 |
| | Full B.C. (k=200) | -6.22 | -6.94 | **-7.14** | -7.13 | 0.49 | 0.68 |
| | B.C. (k=130) | **-6.50** | **-7.00** | -7.03 | **-7.14** | 0.49 | **0.69** |
| w/ Guide | Vanilla (k=1) | -5.47 (+4.6%) | -5.89 (+1.4%) | -6.29 (-0.2%) | -6.31 (+2.3%) | 0.46 (+0.0%) | 0.62 (+0.0%) |
| | Vanilla MC (k=1) | -6.62 (+5.9%) | -7.27 (+8.5%) | -7.74 (+10.4%) | -7.79 (+10.5%) | 0.55 (+7.8%) | 0.65 (+6.6%) |
| | Full B.C. (k=200) | -7.42 (+19.3%) | -7.98 (+15.0%) | -8.25 (+15.5%) | -8.24 (+16.6%) | 0.54 (+10.2%) | 0.76 (+11.8%) |
| | B.C. (k=130) | **-7.52** (+15.7%) | **-8.06** (+15.1%) | **-8.34** (+18.6%) | **-8.40** (+17.6%) | **0.56** (+14.3%) | **0.77** (+11.6%) |

## 6 CONCLUSION

We present MolJO, the joint gradient-based SE(3)-equivariant framework to solve the structure-based molecule optimization problems, which only requires training energy functions as a proxy to predict molecular property, instead of expensive oracle simulations. The general framework further equips gradient-based optimization method with backward correction strategy, offering a flexible trade-off between exploration and exploitation. Experiments show that MolJO is able to improve the binding affinity of molecules by establishing more key interactions and enhance drug-likeness and synthesizability, achieving state-of-the-art performance when benchmarked on CrossDocked2020 (Success Rate 51.3%, Vina Dock -9.05 and SA 0.78), together with $4\times$ improvement compared to gradient-based counterpart and $2\times$ "Me-Better" Ratio as much as other 3D baselines.

## ETHICS STATEMENT

This work is aimed at facilitating structure-based molecule optimziation (SBMO) for drug discovery pipeline. The positive societal impacts include effective design of viable drug candidates. While there is a minimal risk of misuse for generating harmful substances, such risks are mitigated by the need for significant laboratory resources and ethical conduct.

## REPRODUCIBILITY STATEMENT

Reproducibility of this worked is ensured by detailed descriptions of experimental setup in Section 5.1 and implementation in Appendix D.

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

## A OVERVIEW OF BAYESIAN FLOW NETWORKS

In this section, we provide a further explanation of Bayesian Flow Networks (BFNs) that are designed to model the generation of data through a process of message exchange between a "sender" and a "receiver" (Graves et al., 2023). The fundamental elements include the sender distribution for the sender, and input distribution, output distribution, receiver distribution for the receiver.

This process is framed within the context of Bayesian inference, where the *sender distribution* is a factorized distribution $p_S(\mathbf{y}|\mathbf{m}, \alpha_t)$ that introduces noise to each dimension of the data $\mathbf{m}$ and sends it to the receiver, which observes the $\mathbf{y}$ and has access to the noisy channel with accuracy $\alpha$ at timestep $t$, and compares it with its own *receiver distribution* $p_R(\mathbf{y}|\boldsymbol{\theta}, \mathbf{p}; t)$ based on its current belief of the parameters $\boldsymbol{\theta}$, the timestep and any conditional input such as protein pocket $\mathbf{p}$.

The generative process for the receiver begins with a prior distribution (referred to as *input distribution* $p_I(\mathbf{m}|\boldsymbol{\theta}) = \prod_{d=1}^{N} p_I(m^{(d)}|\theta^{(d)})$ that is also factorized for $N$-dimensional data) that defines its initial belief about the data. For continuous data, the prior can be chosen as a Gaussian (Song et al., 2024; Qu et al., 2024), or other distribution such as von Mises distribution (Anonymous, 2024), while for discrete data, the prior is modeled as a uniform categorical distribution. Then, the receiver uses its belief $\boldsymbol{\theta}$ with the help of neural network to model the inter-dependency among dimensions and compute the *output distribution* $p_O(\hat{\mathbf{m}}|\boldsymbol{\theta}, \mathbf{p}; t)$, which represents its estimate of the possible reconstructions of original data $\mathbf{m}$, and is used to construct the receiver distribution (Eq. 3).

The Bayesian update function in Eq. 1 defines how the prior belief $\boldsymbol{\theta}_0$ is updated to the conjugate posterior $\boldsymbol{\theta}_t$. Ideally, the update requires aggregating all possible noisy latents $\mathbf{y}$ from the sender distribution $p_S$, while during the actual generative process, there is only receiver distribution $p_R$, leaving an exposure bias, and different approximations determine different forms of mapping $f$ to the posterior (Eq. 2), showcasing the flexibility in the design space of BFN.

Through the iterative communication between sender and receiver, the receiver progressively updates its belief of the underlying parameters, and training is achieved by minimizing the divergence between the sender and receiver distributions (Eq. 4). This is analogous to the way Bayesian inference works in parameter estimation: as more noisy data $\mathbf{y}$ is observed, the receiver's posterior belief about the data $\mathbf{m}$ becomes increasingly accurate, which implies the reconstruction would be made easier.

## B SENDER DISTRIBUTION FOR DISCRETE DATA

The continuous parameter $\mathbf{z}$ for discrete types $\mathbf{v}$ is updated by observed noisy $\mathbf{y}^v$. Here we briefly introduce how to configure the sender so that $\mathbf{y}^v$ follows a Gaussian as well. For detailed derivation, we refer the readers to Graves et al. (2023).

While true discrete data can be viewed as a sharp one-hot distribution, it can be further relaxed by a factor $\omega \in [0, 1]$ into a Categorical distribution, with the probability $p(k^{(d)}|\mathbf{v}^{(d)}; \omega) = \frac{1-\omega}{K} + \omega \delta_{k^{(d)}\mathbf{v}^{(d)}}$ for $k$ from 1 to $K$ along the $d$-th dimension, where $\delta$ is the Kronecker delta function.

Instead of focusing on the density or sampling once from it, note that the counts $c$ of observing each class in $m$ independent draws follow a multinomial distribution, namely $c \sim \text{Multi}(m, p)$. Dropping the superscripts, Graves et al. (2023) derives the following conclusions:

**Proposition B.1.** *When the number of experiments $m$ is large enough, the frequency approximates its density for class indexed at $k$, i.e. $\lim_{m \to \infty} \frac{c_k}{m} = p(k|\mathbf{v}; \omega)$, following the law of large numbers. Furthermore, by the central limit theorem, it follows that $\frac{c-mp}{\sqrt{mp(1-p)}} \sim \mathcal{N}(0, \mathbf{I})$ when $m \to \infty$.*

**Proposition B.2.** *Denoting $y_k = (c_k - \frac{m}{K}) \ln \xi$ with $\xi = 1 + \frac{\omega K}{1-\omega}$, and $p_S(y_k|\mathbf{v}; \alpha) = \lim_{\omega \to 0} p(y_k|\mathbf{v}; \omega)$ with $\alpha = m\omega^2$, it holds from the change of variables that $p_S(y_k|\mathbf{v}; \alpha) = \mathcal{N}(\alpha(K\delta_{k\mathbf{v}} - 1), \alpha K)$.*

Thus it naturally follows that such noisy $\mathbf{y}^v \sim \mathcal{N}(\mathbf{y}^v|\alpha(K\mathbf{e_v} - \mathbf{1}), \alpha K\mathbf{I})$.

# C  PROOFS

## C.1  PROOF OF GUIDED BAYESIAN UPDATE DISTRIBUTION

**Lemma.** If a random vector $\mathbf{X}$ has probability density $f(\mathbf{x}) \propto \mathcal{N}(\mathbf{x}|\boldsymbol{\mu}, \boldsymbol{\Sigma})e^{\mathbf{c}^{\mathrm{T}}\mathbf{x}}$, where $\mathbf{c}$ is a constant vector with the same dimension as $\mathbf{X}$, then $\mathbf{X} \sim \mathcal{N}(\boldsymbol{\mu} + \boldsymbol{\Sigma}\mathbf{c}, \boldsymbol{\Sigma})$.

*Proof.* We obtain the proof by completing the square as shown below.

$$\log f(\mathbf{x}) = C - \frac{1}{2}(\mathbf{x} - \boldsymbol{\mu})^{\mathrm{T}}\boldsymbol{\Sigma}^{-1}(\mathbf{x} - \boldsymbol{\mu}) + \mathbf{c}^{\mathrm{T}}\mathbf{x}$$
$$= C' - \frac{1}{2}(\mathbf{x} - \boldsymbol{\mu} - \boldsymbol{\Sigma}\mathbf{c})^{\mathrm{T}}\boldsymbol{\Sigma}^{-1}(\mathbf{x} - \boldsymbol{\mu} - \boldsymbol{\Sigma}\mathbf{c}) \tag{16}$$

where $C$ and $C'$ are constant scalars. $\qquad\square$

**Proposition** (4.1). Assuming $\boldsymbol{\mu}_i \sim \mathcal{N}(\boldsymbol{\mu}_\phi, \sigma)$ and $\mathbf{y}_i \sim \mathcal{N}(\mathbf{y}_\phi, \sigma')$ in the original generative process of BFN, we can approximately sample $\boldsymbol{\mu}_i, \mathbf{y}_i$ from the guided transition kernel $\pi(\boldsymbol{\theta}_i|\boldsymbol{\theta}_{i-1})$ according to Eq. 5 and 6.

*Proof.* Under the definition of $\pi$, with the parameters $\boldsymbol{\theta}_i = (\boldsymbol{\mu}_i, \mathbf{z}_i)$ redefined as $(\boldsymbol{\mu}_i, \mathbf{y}_i)$, we have

$$\pi(\boldsymbol{\mu}_i, \mathbf{y}_i|\boldsymbol{\theta}_{i-1}) \propto p_\phi(\boldsymbol{\mu}_i, \mathbf{y}_i|\boldsymbol{\theta}_{i-1})p_E(\boldsymbol{\mu}_i, \mathbf{y}_i) \tag{17}$$

where we have omitted some parentheses and protein pocket condition $\mathbf{p}$ for brevity.

Eq. 14 and 15 guarantee that $\boldsymbol{\mu}_i \sim \mathcal{N}(\boldsymbol{\mu}_\phi, \sigma), \mathbf{y}_i \sim \mathcal{N}(\mathbf{y}_\phi, \sigma')$. Plugging $p_E(\boldsymbol{\mu}_i, \mathbf{y}_i) \propto e^{-E(\boldsymbol{\mu}_i, \mathbf{y}_i, t_i)}$ into Eq. 17, we get

$$\pi(\boldsymbol{\mu}_i, \mathbf{y}_i|\boldsymbol{\theta}_{i-1}) \propto \mathcal{N}(\boldsymbol{\mu}_\phi, \sigma)\mathcal{N}(\mathbf{y}_i|\mathbf{y}_\phi, \sigma')e^{-E(\boldsymbol{\mu}_i, \mathbf{y}_i, t_i)} \tag{18}$$

With $t_i$ fixed, perform a first-order Taylor expansion to $E(\boldsymbol{\mu}, \mathbf{y}, t_i)$ at $(\boldsymbol{\mu}_{i-1}, \mathbf{y}_{i-1})$:

$$E(\boldsymbol{\mu}_i, \mathbf{y}_i, t_i) \approx E(\boldsymbol{\mu}_{i-1}, \mathbf{y}_{i-1}, t_i) - \mathbf{g}_{\boldsymbol{\mu}}^{\mathrm{T}}(\boldsymbol{\mu}_i - \boldsymbol{\mu}_{i-1}) - \mathbf{g}_{\mathbf{y}}^{\mathrm{T}}(\mathbf{y}_i - \mathbf{y}_{i-1}) \tag{19}$$

where gradient $\mathbf{g}_{\boldsymbol{\mu}} = -\nabla_{\boldsymbol{\mu}}E(\boldsymbol{\theta}, t_i)|_{\boldsymbol{\theta}=\boldsymbol{\theta}_{i-1}}$, $\mathbf{g}_{\mathbf{y}} = -\nabla_{\mathbf{y}}E(\boldsymbol{\theta}, t_i)|_{\boldsymbol{\theta}=\boldsymbol{\theta}_{i-1}}$. Substitute it into Eq. 18:

$$\pi(\boldsymbol{\mu}_i, \mathbf{y}_i|\boldsymbol{\theta}_{i-1}) \overset{\mathrm{apx}}{\propto} \mathcal{N}(\boldsymbol{\mu}_i|\boldsymbol{\mu}_\phi, \sigma)\mathcal{N}(\mathbf{y}_i|\mathbf{y}_\phi, \sigma')e^{\mathbf{g}_{\boldsymbol{\mu}}^{\mathrm{T}}\boldsymbol{\mu}_i + \mathbf{g}_{\mathbf{y}}^{\mathrm{T}}\mathbf{y}_i} \tag{20}$$

Eq. 20 together with the lemma above leads to Proposition 4.1.

$\qquad\square$

## C.2  PROOF OF EQUIVARIANCE

**Proposition** (4.4). The guided sampling process preserves SE(3)-equivariance when $\boldsymbol{\Phi}$ is SE(3)-equivariant, if the energy function $E(\boldsymbol{\theta}, \mathbf{p}, t)$ is also parameterized with an SE(3)-equivariant neural network, and the complex is shifted to the space where the protein's Center of Mass (CoM) is zero.

*Proof.* Following Schneuing et al. (2022), once the complex is moved so that the pocket is centered at the origin (*i.e.* zero CoM), translation equivariance becomes irrelevant and only O(3)-equivariance needs to be satisfied.

For any orthogonal matrix $\mathbf{R} \in \mathbb{R}^{3\times 3}$ such that $\mathbf{R}^{\top}\mathbf{R} = \mathbf{I}$, it is easy to see that the prior $\boldsymbol{\mu}_0 = \mathbf{0}$ is O(3)-invariant. Given that $\hat{\mathbf{x}} \sim p_O(\hat{\mathbf{x}} \mid \boldsymbol{\Phi}(\boldsymbol{\theta}, \mathbf{p}, t))$ and the equivariance of $\boldsymbol{\Phi}$, it suffices to prove the invariant likelihood for the transition kernel to complete the proof.

Given the parameterization of pretrained energy function $E(\boldsymbol{\theta}, \mathbf{p}, t)$ is SE(3)-equivariant, then the gradient $\mathbf{g}_{\boldsymbol{\mu}}(\boldsymbol{\theta}) = -\nabla_{\boldsymbol{\mu}}E(\boldsymbol{\theta}, \mathbf{p}, t_i)$ is also equivariant according to Bao et al. (2022).

Without loss of generality, we consider the guided transition density for $i \leq k$, which simplifies to

$$\pi(\boldsymbol{\mu}_i \mid \boldsymbol{\mu}_{i-1}, \mathbf{y}_{i-1}, \mathbf{p})$$
$$= \mathcal{N}(\boldsymbol{\mu}_i \mid \gamma_i\boldsymbol{\Phi}(\boldsymbol{\mu}_{i-1}, \mathbf{y}_{i-1}, \mathbf{p}) + \gamma_i(1 - \gamma_i)\mathbf{g}_{\boldsymbol{\mu}}(\boldsymbol{\mu}_{i-1}, \mathbf{y}_{i-1}, \mathbf{p}, t_{i-1}), \gamma_i(1 - \gamma_i)\mathbf{I})$$

where $\gamma_i \stackrel{\text{def}}{:=} \frac{\beta(t_i)}{1+\beta(t_i)}$.

Then we can prove that it is O(3)-invariant:

$$
\begin{aligned}
&\pi(\mathbf{R}\boldsymbol{\mu}_i \mid \mathbf{R}\boldsymbol{\mu}_{i-1}, \mathbf{y}_{i-1}, \mathbf{R}\mathbf{p}) \\
&= \mathcal{N}(\mathbf{R}\boldsymbol{\mu}_i \mid \gamma_i\boldsymbol{\Phi}(\mathbf{R}\boldsymbol{\mu}_{i-1}, \mathbf{y}_{i-1}, \mathbf{R}\mathbf{p}) + \gamma_i(1-\gamma_i)\mathbf{g}_{\boldsymbol{\mu}}(\mathbf{R}\boldsymbol{\mu}_{i-1}, \mathbf{y}_{i-1}, \mathbf{R}\mathbf{p}, t_{i-1}), \gamma_i(1-\gamma_i)\mathbf{I}) \\
&= \mathcal{N}(\mathbf{R}\boldsymbol{\mu}_i \mid \gamma_i\boldsymbol{\Phi}(\mathbf{R}\boldsymbol{\mu}_{i-1}, \mathbf{y}_{i-1}, \mathbf{R}\mathbf{p}) + \gamma_i(1-\gamma_i)\mathbf{R}\mathbf{g}_{\boldsymbol{\mu}}(\boldsymbol{\mu}_{i-1}, \mathbf{y}_{i-1}, \mathbf{p}, t_{i-1}), \gamma_i(1-\gamma_i)\mathbf{I}) \\
&\qquad\qquad\qquad\qquad\qquad\qquad\qquad\qquad\qquad\qquad\qquad\qquad (\text{equivariance of } \mathbf{g}_{\boldsymbol{\mu}}) \\
&= \mathcal{N}(\mathbf{R}\boldsymbol{\mu}_i \mid \gamma_i\mathbf{R}\boldsymbol{\Phi}(\boldsymbol{\mu}_{i-1}, \mathbf{y}_{i-1}, \mathbf{p}) + \gamma_i(1-\gamma_i)\mathbf{R}\mathbf{g}_{\boldsymbol{\mu}}(\boldsymbol{\mu}_{i-1}, \mathbf{y}_{i-1}, \mathbf{p}, t_{i-1}), \gamma_i(1-\gamma_i)\mathbf{I}) \\
&\qquad\qquad\qquad\qquad\qquad\qquad\qquad\qquad\qquad\qquad\qquad\qquad (\text{equivariance of } \boldsymbol{\Phi}) \\
&= \mathcal{N}(\boldsymbol{\mu}_i \mid \gamma_i\boldsymbol{\Phi}(\boldsymbol{\mu}_{i-1}, \mathbf{y}_{i-1}, \mathbf{p}) + \gamma_i(1-\gamma_i)\mathbf{g}_{\boldsymbol{\mu}}(\boldsymbol{\mu}_{i-1}, \mathbf{y}_{i-1}, \mathbf{p}, t_{i-1}), \gamma_i(1-\gamma_i)\mathbf{I}) \\
&\qquad\qquad\qquad\qquad\qquad\qquad\qquad\qquad\qquad\qquad\qquad (\text{equivariance of isotropic Gaussian}) \\
&= \pi(\boldsymbol{\mu}_i \mid \boldsymbol{\mu}_{i-1}, \mathbf{y}_{i-1}, \mathbf{p})
\end{aligned}
$$

It also applies to cases where $i > k$, for we can recurrently view the starting point of backward corrected history $\boldsymbol{\mu}_{i-k}$ as the new O(3)-invariant prior $\boldsymbol{\mu}_0$ and iteratively make the above derivation.
$\square$

## D   IMPLEMENTATION DETAILS

### D.1   MODEL DETAILS

**Backbone.**   Our BFN backbone follows that of MolCRAFT (Qu et al., 2024), and we conduct optimization during sampling on the pretrained checkpoint without finetuning.

**Training Property Regressors.**   In order for a differentiable oracle function, we additionally train the energy function based on the molecules and their properties (Vina Score, QED, SA) in CrossDocked dataset (Francoeur et al., 2020) by minimizing the squared loss for property $c$ over the data distribution $p_{\text{data}}$:

$$
L = \mathbb{E}_{p_{\text{data}}}|E(\boldsymbol{\theta}, \mathbf{p}, t) - c|^2 \tag{21}
$$

The input parameters to all energy functions belong to the parameter space defined by $\beta_1 = 1.5$ for atom types, $\sigma_1 = 0.03$ for atom coordinates, $n = 1000$ discrete steps. The energy network is parameterized with the same model architecture as TargetDiff (Guan et al., 2022), *i.e.* kNN graphs with $k = 32$, $N = 9$ layers with $d = 128$ hidden dimension, 16-headed attention, and the same featurization, *i.e.* protein atoms (H, C, N, O, S, Se) and ligand atoms (C, N, O, F, P, S, Cl). For training, Adam optimizer is adopted with learning rate 0.005, batch size is set to 8. The training takes less than 8 hours on a single RTX 3090 and converges within 5 epochs.

**Sampling.**   To sample via guided Bayesian flow, we set the sample steps to 200, and the guidance scale to 50. For the combination different objectives, we merely take an average of different gradients.

### D.2   TASK DETAILS

**R-group optimization.**   Cases of lead optimization involve retaining the scaffold while redesigning the remaining R-groups, usually when the scaffold forms desirable interactions with the protein and anchors the binding mode, and the remaining parts need further modifications to secure this pattern and enhance binding affinity. Following Polykovskiy et al. (2020), we employ RDKit for fragmentization and atom annotation with R-group or Bemis-Murcko scaffold.

**Scaffold hopping.**   Different from R-group design, scaffold hopping means to redesign the scaffold for a given molecule while keeping its core functional groups, for example to overcome the patent protection for a known drug molecule while retaining pharmaceutical activity. This is a technically more challenging task for generative models, for the missing parts they need to fulfill are generally larger than those in R-group design, and the hopping is usually subject to more chemical constraints. We construct scaffold hopping as a dual problem to R-group optimization with Bemis-Murcko scaffolding annotation, although it does not need to be so.

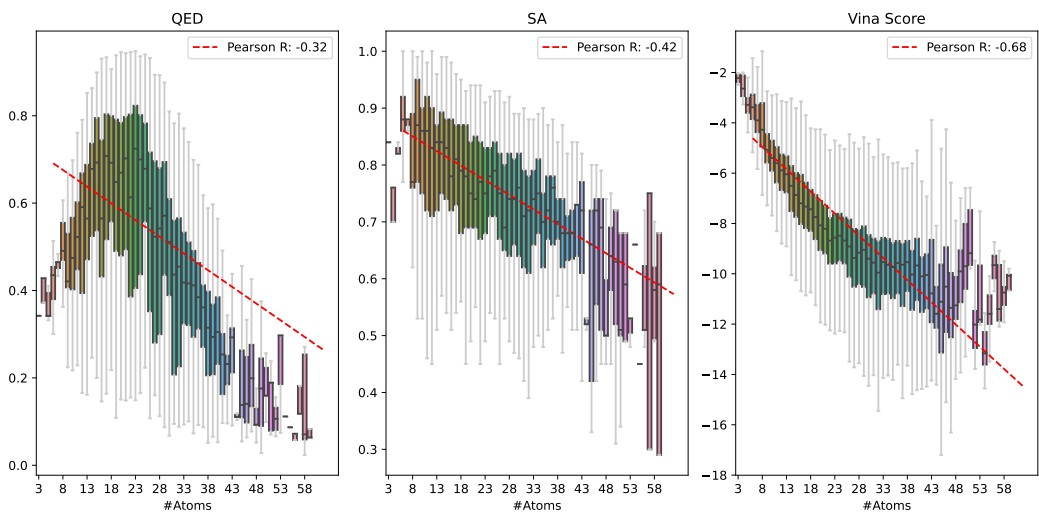

Figure 6: Distribution of molecular properties (QED, SA, Vina Score) over the number of atoms for CrossDocked2020. For each size, the mean and error bars are shown in the boxplot.

Table 5: Molecular properties under different sizes, where *Ref Size* denotes 23 atoms on average, and *Large Size* around 30 atoms. Top 2 results are highlighted with **bold text** and underlined text.

| Methods | | Vina Score (↓) | | Vina Min (↓) | | Vina Dock (↓) | | QED (↑) | SA (↑) | Div (↑) | Success |
|---|---|---|---|---|---|---|---|---|---|---|---|
| | | Avg. | Med. | Avg. | Med. | Avg. | Med. | Avg. | Avg. | Avg. | Rate (↑) |
| Ref Size | DecompDiff | -5.19 | -5.27 | -6.03 | -6.00 | -7.03 | -7.16 | 0.51 | 0.66 | **0.73** | 14.9% |
| | DecompOpt | -5.75 | -5.97 | -6.58 | -6.70 | -7.63 | -8.02 | **0.56** | 0.73 | 0.63 | 39.4% |
| | MolCRAFT | -6.59 | -7.04 | -7.27 | -7.26 | -7.92 | -8.01 | 0.50 | 0.69 | 0.72 | 26.0% |
| | MolJO | -7.52 | -8.02 | -8.33 | -8.34 | -9.05 | -9.13 | **0.56** | **0.78** | 0.66 | 51.3% |
| Large Size | DecompDiff | -5.67 | -6.04 | -7.04 | -7.09 | -8.39 | -8.43 | 0.45 | 0.61 | 0.68 | 24.5% |
| | DecompOpt | -5.87 | -6.81 | -7.35 | -7.72 | -8.98 | -9.01 | 0.48 | 0.65 | 0.60 | 52.5% |
| | MolCRAFT | -6.61 | -8.14 | -8.14 | -8.42 | -9.25 | -9.20 | 0.46 | 0.62 | 0.61 | 36.6% |
| | MolJO | **-7.93** | **-9.26** | **-9.47** | **-9.73** | **-10.53** | **-10.48** | 0.50 | 0.72 | 0.57 | **64.2%** |

# E    EFFECT OF MOLECULAR SIZE ON PROPERTIES

The size of molecules are found with a notable impact over molecular properties including Vina affinities (Qu et al., 2024). We quantify the relationship and plot the distribution of molecular properties w.r.t. the number of atoms with the Pearson correlation coefficient in Figure 6. It is not surprising to see a non-negligible correlation between properties and molecular sizes, since the size of molecules typically constrain its possibility over the chemical space. To ensure a fair comparison, we stick to the molecular space with similar size to the reference. For further comparison among different model variants, we report the molecular properties under different sizes in Table 5. Results show that our method consistently achieves the highest success rate, demonstrating its robust optimization ability even in an Out-of-Distribution (OOD) scenario.

# F    FULL OPTIMIZATION RESULTS

**Baselines.**    We provide a detailed description of all baselines here:

- **AR** (Luo et al., 2021) uses MCMC sampling to reconstruct a molecule atom-by-atom given voxel-wise densities.

- **GraphBP** (Liu et al., 2022) is an autoregressive atom-based model that uses normalizing flow and encodes the context to preserve 3D geometric equivariance.

- **Pocket2Mol** (Peng et al., 2022) generates one atom with bond at a time via an E(3)-equivariant network. It predicts frontier atoms to expand, alleviating the efficiency problem in sampling.

- **FLAG** (Zhang et al., 2023) is a fragment-based model that assembles the generated fragments via predicted coordinates and torsion angles.

- **DiffSBDD** (Schneuing et al., 2022) constructs an equivariant continuous diffusion for full-atom generation given pocket information, and applies Gaussian noise to both continuous atom coordinates and discrete atom types.

- **TargetDiff** (Guan et al., 2022) adopts a continuous-discrete diffusion approach that treats each modality via corresponding diffusion process, achieving better performance than continuous diffusion such as DiffSBDD.

- **DecompDiff** (Guan et al., 2023) decomposes the molecules into contact arms and linking scaffolds, and utilizes such chemical priors in the diffusion process.

- **IPDiff** (Huang et al., 2024) pretrains an affinity predictor, and utilizes this predictor to extract features that augments the conditioning of diffusion generative process.

- **MolCRAFT** (Qu et al., 2024) employs Bayesian Flow Network for molecular design with an advanced sampling strategy, showing notable improvement upon diffusion counterparts.

- **AutoGrow4** (Spiegel & Durrant, 2020) is an evolutionary algorithm that uses genetic algorithm to optimize 1D SMILES, with additional docking simulation. Starting from the initial seed molecule, AutoGrow4 iteratively conducts mutations and crossovers, then makes oracle calls for docking feedback, and keeps the top-scoring molecules in the end.

- **RGA** (Fu et al., 2022) is built on top of AutoGrow4, and utilizes a pocket-aware RL-trained policy to suppress its random walking behavior in traversing the molecular space.

- **DecompOpt** (Zhou et al., 2024) trains a conditional generative model on decomposed fragments besides binding pocket, following the style of DecompDiff. The optimization is done by iteratively re-sampling in the 3D diffusion latent space given the top $K$ arms ranked by oracle functions as updated fragment condition input.

- **TAGMol** (Dorna et al., 2024) exerts gradient-based property guidance over the pretrained TargetDiff backbone, and the gradient is enabled only in the continuous diffusion process for coordinates.

**Metrics.** Besides the common evaluation metrics such as binding affinities calculated by Autodock Vina (Eberhardt et al., 2021) and QED, SA by RDKit, we elaborate other metrics as follows:

- **Diversity** measures the diversity of generated molecules for each binding site. Following SBDD convention (Luo et al., 2021), it is based on Tanimoto similarity over Morgan fingerprints, and averaged on 100 test proteins.

- **Connected Ratio** is the ratio of complete molecules among all generated molecules, *i.e.* with only one connected component.

- **Lipinski** enumerates Lipinski rule of five (Lipinski et al., 1997) and checks how many rules are satisfied. These rules are typically seen as empirical reference that helps to predict whether the molecule is likely to be orally bioavailable.

- **Key Interaction**, *i.e.* key non-covalent interactions formed between molecules and protein binding sites as an in-depth measure for binding modes, including $\pi$ interactions, hydrogen bonds (donor and acceptor), salt bridges and hydrophobic interactions calculated by Schrödinger Glide (Halgren et al., 2004).

- **Strain Energy** measures the internal energy of generated poses, serving as an indicator of pose quality as proposed by Harris et al. (2023).

- **Steric Clash** calculates the number of clashes between generated ligand and protein surface, where clashing means the distance of ligand and protein atoms are within a certain threshold. This reveals the stability of complex to some extend, yet it does not strictly mean violation of physical constraint since the protein is not that rigid and might also go through spatial rearrangement upon binding, as noted by Harris et al. (2023).

- **Redocking RMSD** reports the percentage of molecules with an RMSD between generated and Vina redocked poses lie within the range of 2Å, which suggests the binding mode remains consistent after redocking.

### F.1    MOLECULE OPTIMIZATION

**Overall Distributions.**    We additionally report the property distributions for Vina Score, SA, and QED in Figure 15, 13, 14, respectively, demonstrating the efficacy of our proposed method in optimizing a number of objectives for "me-better" drug candidates.

**Affinity Analysis.**    We present the tail distribution of Vina affinities in Table 6, demonstrating that our method not only excels in optimizing overall performance as shown in Figure 3, but also enhances the quality of the best possible binders.

To better understand the enhanced binding affinites, we further analyze the distribution of non-covalent interactions that are known to play an important role in stabilizing protein-ligand complexes. Figure 7 demonstrates that the improved affinity results are achieved by forming a greater number of hydrophobic interactions, more hydrogen bond acceptors as well as $\pi$ interactions.

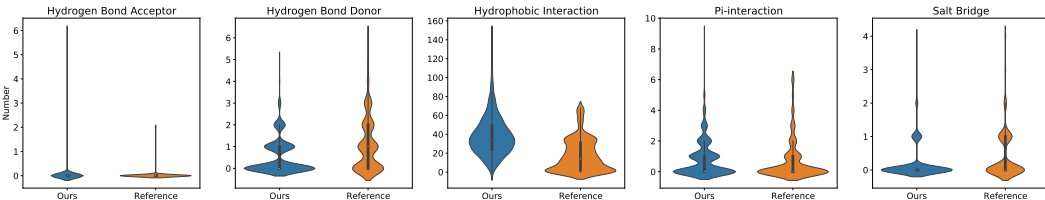

Figure 7: Non-covalent interaction distributions of reference and optimized molecules.

Table 6: Tail distribution of Vina affinities.

|  | Vina Score 5% | Vina Min 5% | Vina Dock 5% |
|---|---|---|---|
| Reference | -9.98 | -9.93 | -10.62 |
| AR | -10.05 | -10.33 | -10.56 |
| Pocket2Mol | -10.47 | -11.77 | -12.36 |
| TargetDiff | -11.10 | -11.57 | -11.89 |
| DecompDiff | -10.04 | -10.96 | -11.77 |
| IPDiff | -12.98 | -13.40 | -13.63 |
| MolCRAFT | -12.14 | -12.34 | -12.58 |
| DecompOpt | -10.78 | -11.70 | -12.73 |
| TAGMol | -13.15 | -13.50 | -13.67 |
| MolJO | **-13.59** | **-13.90** | **-14.18** |

**Combination of Objectives.**    In Table 7, we report the results for an exhausted combination of different objectives under the unconstrained setting, where 1000 molecules are sampled in total. It can be seen that combining two objectives yields nearly the best optimized performances for each objective, with the choice of Affinity + SA even displaying improvement in QED. However, from the QED + SA setting, we observe a negative impact on binding affinity. It is possible that too high a requirement of QED and SA further constrains the chemical space for drug candidates, limiting the types of potential interactions with protein surfaces. When it comes to all objectives, MolJO achieves balanced optimization results, *i.e.* satisfactory QED and SA comparable to single objective optimization or the combination of two, and enhanced affinities compared with the results without affinity optimization, though slightly inferior to the best possible affinity optimization results. This might stem from QED + SA problems described above, suggesting a careful handling of these two objectives. In this regard, we simply choose Affinity + SA objectives in all our main experiments for a clear demonstration of our optimization ability.

For a better understanding of the correlation between objectives, we plot the pairwise relationships for the molecules in the training set in Figure 8, and calculate the Spearman's rank coefficient of

correlation $\rho$. The Spearman $\rho$ is 0.41 between SA and QED, and it is reasonable to see such a positive correlation between SA and QED, since these are both indicators of drug-likeness with certain focus and thus alternative to some extent. This aligns with our findings that adopting the Affinity + SA objectives can also benefit QED, and justifies our choice of optimization objectives in this sense. Moreover, although there is also a slightly positive correlation between Vina Score and SA ($\rho = 0.33$), meaning that it is nontrivial to simultaneously optimize both properties, our method succeeds in finding the best balanced combination of properties, demonstrating the superiority of joint optimziation compared with TAGMol.

Table 7: Combinations of different objectives. Top-2 results are highlighted in **bold** and underlined, respectively.

| Objective | Vina Score ($\downarrow$) Avg. | Med. | Vina Min ($\downarrow$) Avg. | Med. | QED ($\uparrow$) | SA ($\uparrow$) | Connected ($\uparrow$) |
|---|---|---|---|---|---|---|---|
| Affinity | **-7.74** | -7.96 | **-8.21** | -8.19 | 0.52 | 0.68 | 0.87 |
| QED | -6.84 | -7.32 | -7.54 | -7.65 | **0.66** | 0.70 | 0.99 |
| SA | -6.25 | -7.24 | -7.48 | -7.65 | 0.57 | **0.78** | 0.97 |
| QED+SA | -6.55 | -7.23 | -7.38 | -7.52 | 0.65 | 0.74 | 0.99 |
| Affinity+QED | -7.46 | **-8.04** | -8.18 | -8.20 | 0.64 | 0.67 | 0.98 |
| Affinity+SA | -7.08 | -7.88 | -8.05 | **-8.21** | 0.57 | 0.75 | 0.97 |
| All | -7.09 | -7.47 | -7.79 | -7.76 | 0.62 | 0.73 | 0.98 |

## F.2 ABLATION STUDIES

Table 8: Ablation studies of joint optimization for atom types and coordinates, where w/o type means the gradient is disabled for types. Top 2 results are highlighted with **bold** and underlined text.

| Objective | Methods | Vina Score ($\downarrow$) Avg. | Med. | Vina Min ($\downarrow$) Avg. | Med. | QED ($\uparrow$) | SA ($\uparrow$) |
|---|---|---|---|---|---|---|---|
| Affinity | Ours | **-7.74** | **-7.96** | **-8.21** | **-8.19** | 0.52 | 0.68 |
| | w/o type | -7.13 | -7.58 | -7.82 | -7.80 | 0.50 | 0.66 |
| | w/o coord | -6.61 | -7.23 | -7.42 | -7.53 | 0.52 | 0.71 |
| QED | Ours | -6.84 | -7.32 | -7.54 | -7.65 | **0.66** | 0.70 |
| | w/o type | -6.41 | -7.03 | -7.20 | -7.26 | 0.52 | 0.67 |
| | w/o coord | -6.50 | -7.20 | -7.44 | -7.42 | 0.65 | 0.70 |
| SA | Ours | -6.25 | -7.24 | -7.48 | -7.65 | 0.57 | **0.78** |
| | w/o type | -6.29 | -6.85 | -7.07 | -7.09 | 0.51 | 0.70 |
| | w/o coord | -6.71 | -7.22 | -7.60 | -7.60 | 0.57 | 0.77 |

**Effect of Joint Guidance.**  Table 8 shows the effectiveness of joint guidance over coordinates or types. Utilizing gradients to guide both data modalities is consistently better than applying single gradient only, since the energy landscape of a molecular system is a function of both the atom coordinates and the types. Lack of direct control over either modality can lead to suboptimal performance due to not efficiently exploring the chemical space where certain atomic types naturally pair with specific spatial arrangements. Specifically, it can be seen that for affinities, the optimization is closely related to coordinates, while for drug-like properties, simply propagating gradients over coordinates displays no improvement at all. This validates our choice of finding appropriate guidance form jointly, and a single coordinate guidance would be insufficient for generating desirable molecules.

**Effect of Backward Correction.**  We conduct ablation studies regarding the proposed backward correction strategy. `w/o Correction` denotes sampling $\boldsymbol{\theta}_i$ according to Eq. 11. Figure 9 shows that increasing the steps $k$ in Eq. 13 that have been corrected backward boosts the optimization performance once sufficient past steps are corrected for optimization.

It can be inferred that sampling $p_\phi(\boldsymbol{\theta}_i | \boldsymbol{\theta}_{i-1}, \boldsymbol{\theta}_{i-k})$ up until $\boldsymbol{\theta}_n$ results in a chain of parameters $\{\boldsymbol{\theta}_{T_i}\}_{i=0}^{\lfloor n/k \rfloor}$, where $T_i = ik + (n \mod k)$, and $\boldsymbol{\theta}_i \sim p_\phi(\boldsymbol{\theta}_i, | \boldsymbol{\theta}_{i-1}, \boldsymbol{\theta}_0)$ when $i \leq (n \mod k)$.

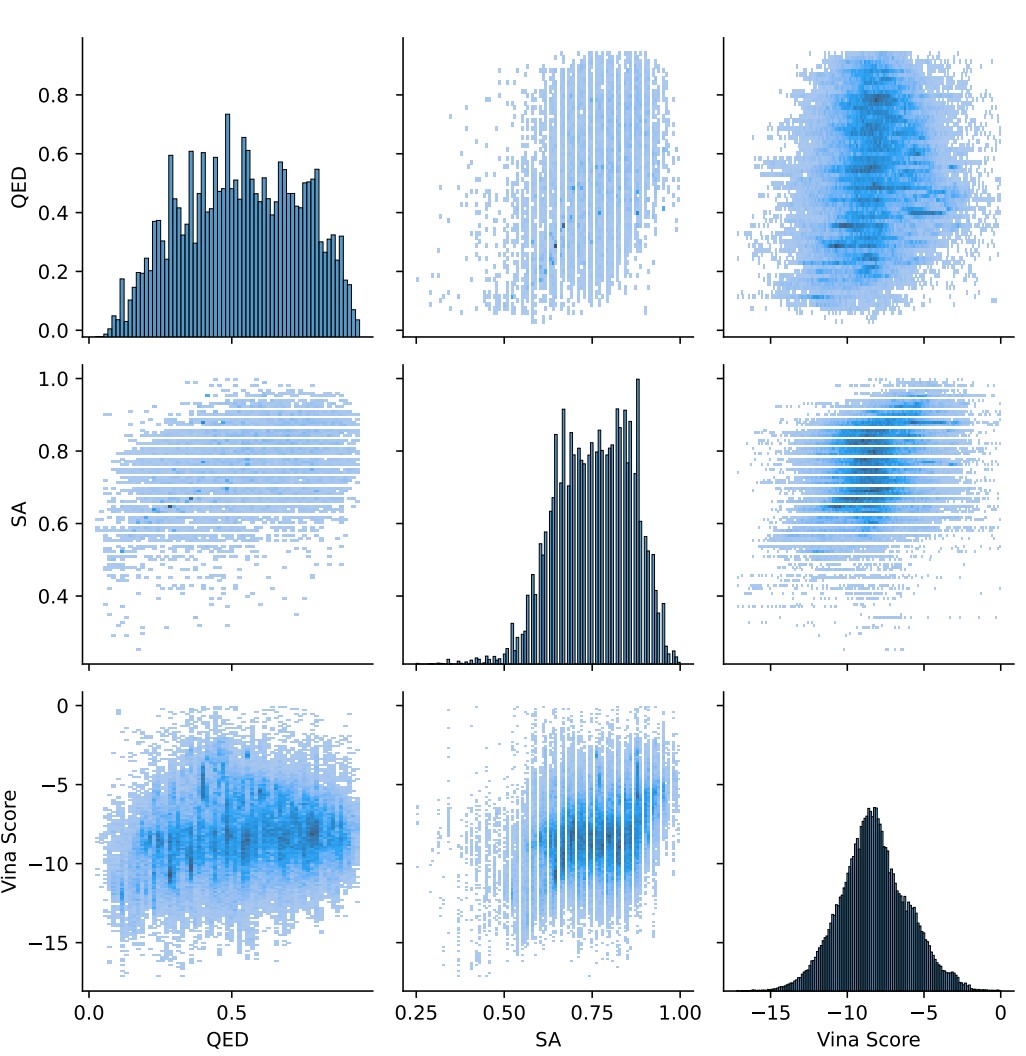

Figure 8: Pairwise correlation of different properties. On the diagonal are histograms showing single property distributions on CrossDocked2020.

Smaller number $k$ of corrected steps moves the starting point $\boldsymbol{\theta}_{T_0}$ closer to $\boldsymbol{\theta}_0$ and sees more updates along the chain. We observe that when $k$ is too small, the sampling process tends to suffer from error accumulation instead of error correction due to stochasticity. Once $k$ is larger than 50, the process is better balanced in exploiting the shortcut (*i.e.* interval $k$) and exploring the stochasticity to reduce approximation errors via a few updates (*i.e.* $\lfloor n/k \rfloor$). The final $k$ is set to 130.

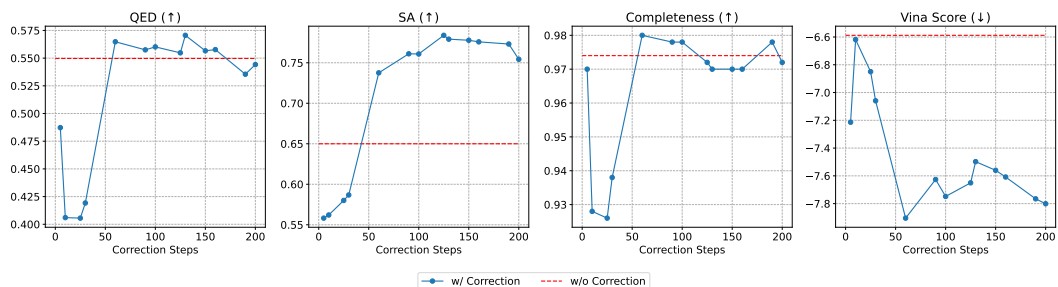

Figure 9: Ablation study of backward correction. Correction Step on the x-axis means the length of history $k$, and w/o Correction means vanilla update ($k = 1$) with a Monte-Carlo estimate of $\mathbf{y}$.

**Effect of Scales.**   We conduct a grid search of guidance scales, and report the full results of ablation studies on different guiding scales within the range $\{0.1, 1, 10, 20, 50, 100\}$ for different objectives (Affinity, QED, SA) in Table 9, where 10 molecules are sampled for each of the 100 test proteins.

For binding affinity, the optimization performance steadily improves with increasing scales, but the ratio of complete molecules significantly decreases when the scale is greater than 50.

For QED and SA, MolJO achieves best results when the scale is around 20 and 50.

In order to maintain the comparability with molecules without guidance, we stick to the scale range where the connected ratio remains acceptable, and therefore set the guidance scale to 50 for all our experiments.

Table 9: Full ablation studies on different guiding scales for different objectives. Top-1 values are highlighted in **bold**.

| Objective | Scale | Vina Score ($\downarrow$) Avg. | Med. | Vina Min ($\downarrow$) Avg. | Med. | QED ($\uparrow$) | SA ($\uparrow$) | Connected ($\uparrow$) |
|---|---|---|---|---|---|---|---|---|
| Affinity | 0.1 | -6.28 | -6.98 | -7.17 | -7.25 | 0.50 | 0.70 | 0.96 |
| | 1 | -6.24 | -7.01 | -7.27 | -7.29 | 0.50 | 0.69 | 0.96 |
| | 10 | -6.69 | -7.46 | -7.46 | -7.67 | 0.51 | 0.70 | 0.97 |
| | 20 | -7.03 | -7.87 | -7.84 | -8.08 | 0.51 | 0.70 | 0.98 |
| | 50 | -7.64 | -8.38 | -8.39 | -8.64 | 0.53 | 0.68 | 0.90 |
| | 100 | **-9.33** | **-9.55** | **-9.87** | **-9.85** | 0.55 | 0.63 | 0.55 |
| QED | 0.1 | -6.03 | -6.92 | -7.10 | -7.19 | 0.51 | 0.70 | 0.97 |
| | 1 | -6.24 | -7.09 | -7.31 | -7.31 | 0.56 | 0.71 | 0.96 |
| | 10 | -6.12 | -7.07 | -7.29 | -7.41 | **0.66** | 0.71 | 0.98 |
| | 20 | -6.33 | -7.23 | -7.34 | -7.64 | **0.66** | 0.69 | 0.98 |
| | 50 | -6.84 | -7.32 | -7.54 | -7.65 | **0.66** | 0.70 | **0.99** |
| | 100 | -6.25 | -6.83 | -7.02 | -7.10 | 0.62 | 0.60 | 0.95 |
| SA | 0.1 | -6.30 | -7.11 | -7.19 | -7.29 | 0.50 | 0.70 | 0.96 |
| | 1 | -6.17 | -7.16 | -7.36 | -7.37 | 0.52 | 0.73 | 0.97 |
| | 10 | -5.87 | -7.23 | -7.39 | -7.72 | 0.57 | 0.78 | 0.98 |
| | 20 | -6.14 | -7.24 | -7.49 | -7.72 | 0.56 | **0.79** | 0.98 |
| | 50 | -6.38 | -7.29 | -7.86 | -7.77 | 0.54 | **0.79** | **0.99** |
| | 100 | -6.08 | -7.36 | -7.51 | -7.85 | 0.54 | 0.78 | 0.98 |

# G  EVALUATION OF MOLECULAR CONFORMATION

**PoseCheck Analysis.**    To measure the quality of generated ligand poses, we further employ PoseCheck Harris et al. (2023) to calculate the Strain Energy (**Energy**) of molecular conformations and Steric Clashes (**Clash**) w.r.t. the protein atoms in Figure 10 and 11, respectively.

Our proposed MolJO not only significantly outperforms the other optimization baselines in both Energy and Clash, but also shows competitive results with strong-performing generative models, in which Pocket2Mol achieves lower strain energy via generating structures with fewer rotatable bonds as noted by Harris et al. (2023), and fragment-based model FLAG directly incorporates rigid fragments in its generation. As for clashes, we achieve the best results in non-autoregressive methods.

Notably, IPDiff ranks the least in Strain Energy and displays severely strained structures despite its strong performance in binding affinities. This arguably suggests that directing utlizing pretrained binding affinity predictor as feature extractor might result in spurious correlated features, even harming the molecule generation.

**RMSD Distribution.**    We report the ratio of redocking **RMSD** below 2Å between generated poses and Vina docked poses to reveal the agreement of binding mode. Due to issues of poses generate by Autodock, not all pose pairs are available for calculating symmetry-corrected RMSD, where we report the non-corrected RMSD instead to make sure that all samples are faithfully evaluated. As shown in Figure 12, the optimization methods all display a tendency towards generating a few outliers, which might be attributed to the somewhat out-of-distribution (OOD) nature of optimization that seeks to shift the original distribution. Among all, DecompOpt generates the most severe outliers with RMSD as high as 160.7 Å, and its unsatifactory performance is also suggested by the lowest ratio of RMSD $< 2$Å (24.3%), while for gradient-based TAGMol and our method, it only has a negligible impact and the ratio is generally more favorable.

**Overall Conformation Quality and Validity.**    The overall results in Table 10 show that our gradient-based method actually improves upon the conformation stability of backbone in terms of energy and clash, demonstrating its ability to faithfully model the chemical environment of protein-ligand complexes, while DecompOpt generates heavily strained structures similar to DecompDiff, and TAGMol ends up with even worse energy than its backbone TargetDiff. Moreover, from the perspective of validity reflected by **Connected Ratio**, the optimization efficiency of RGA and DecompOpt is relatively low as suggested by the ratio of successfully optimized molecules.

**Ring Size.**    For a comprehensive understanding of the effect of property guidance, we additionally report the distribution of ring sizes in Table 11, showing that the gradient-based property guidance generally favors more rings, but our result still lies within a reasonable range, and even improves upon the ratio of 4-membered rings.

Table 10: Summary of conformation stability results. Energy, Clash are calculated by PoseCheck. Connected is the ratio of successfully generated valid and connected molecules.

|  | Energy Med. ($\downarrow$) | Clash Avg. ($\downarrow$) | RMSD $< 2$Å ($\uparrow$) | Connected ($\uparrow$)[1] |
|---|---|---|---|---|
| Reference | 114 | 5.46 | 34.0% | 100% |
| AR | 608 | **4.18** | 36.5% | 93.5% |
| Pocket2Mol | 186 | 6.22 | 31.3% | 96.3% |
| FLAG | 396 | 40.83 | 8.2% | 97.1% |
| TargetDiff | 1208 | 10.67 | 31.0% | 90.4% |
| DecompDiff | 983 | 14.23 | 25.1% | 72.0% |
| IPDiff | 5861 | 10.31 | 17.9% | 90.1% |
| MolCRAFT | 196 | 6.91 | 42.4% | 96.7% |
| RGA | - | - | - | 52.2% |
| DecompOpt | 861 | 16.6 | 24.3% | 2.64% |
| TAGMol | 2058 | 7.41 | 37.2% | 92.0% |
| MolJO | **163** | 6.72 | **43.5%** | **97.3%** |

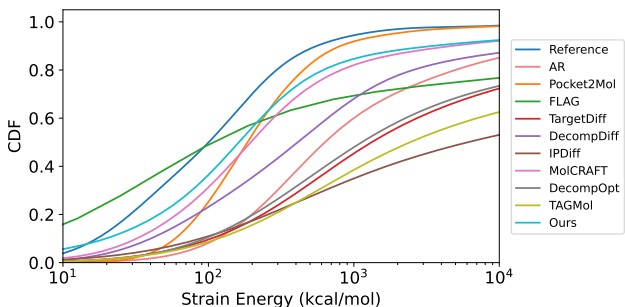

Figure 10: Cummulative density function (CDF) for strain energy distributions of generated molecules and reference molecules.

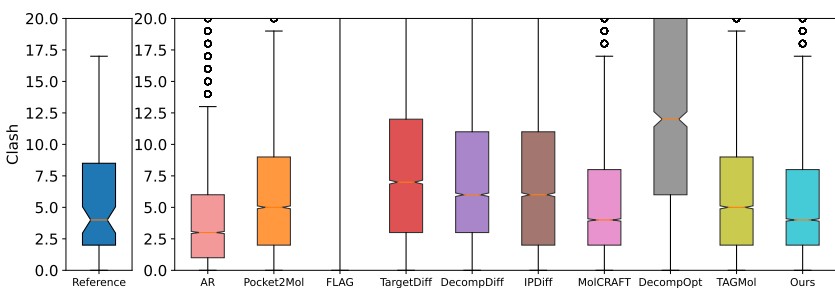

Figure 11: Box plot for clash distributions of generated molecules and reference molecules.

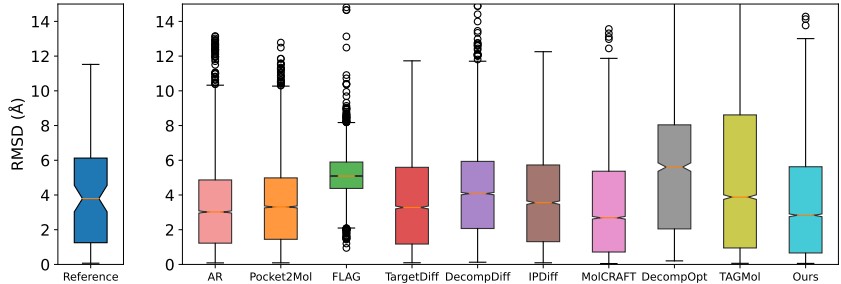

Figure 12: Boxplot for RMSD distributions of generated molecules and reference molecules.

## H  INFERENCE TIME

We report the time cost in Table 12 for optimization baselines in the table below, which is calculated as the time for sampling a batch of 5 molecules on a single NVIDIA RTX 3090 GPU, averaged over 10 randomly selected test proteins.

## I  MORE RELATED WORKS

**Molecule Optimization**    As an alternative to target-aware generative modeling of 3D molecules, the optimization methods are goal-directed, obtain desired ligands usually by searching in the drug-like chemical space guided by property signals (Bilodeau et al., 2022; Du et al., 2024). General optimization algorithms were originally designed for ligand-based drug design (LBDD) and optimize

---

[1]Connected ratio of DecompOpt and RGA is calculated based on the optimization results of all rounds provided by the authors. For each of the $N$ rounds, a total of $k$ molecules ought to be generated, thus we divide the total number of optimized molecules for n pockets by $N \times k \times n$.

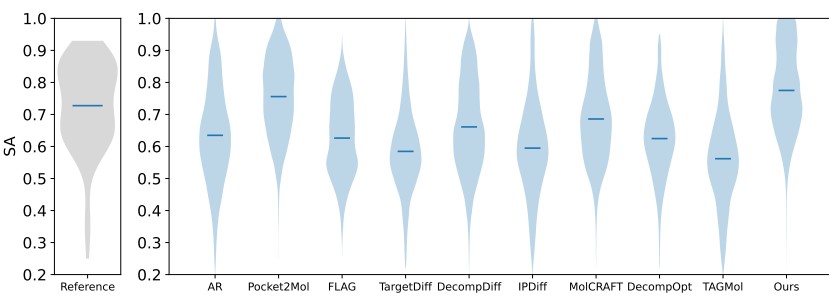

Figure 13: Violin plot for SA distributions of generated molecules and reference molecules.

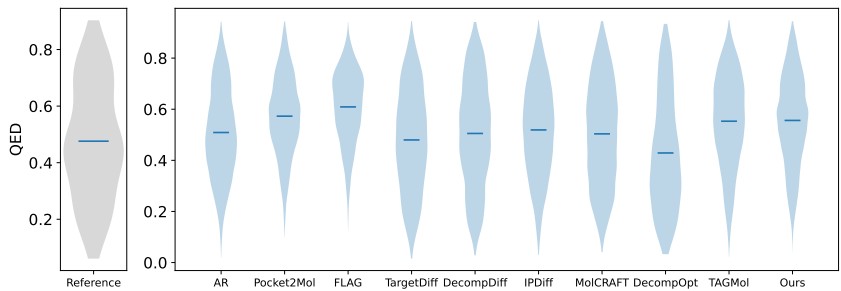

Figure 14: Violin plot for QED distributions of generated molecules and reference molecules.

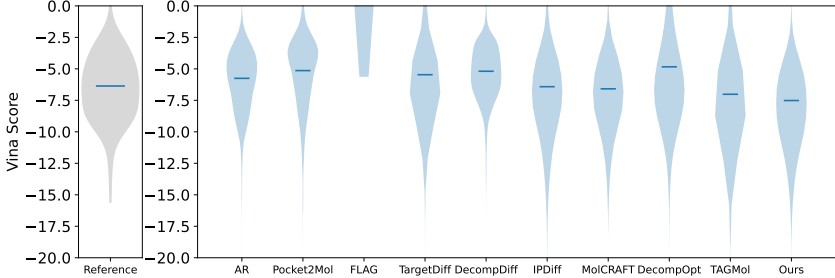

Figure 15: Violin plot for Vina Score distributions of generated molecules and reference molecules.

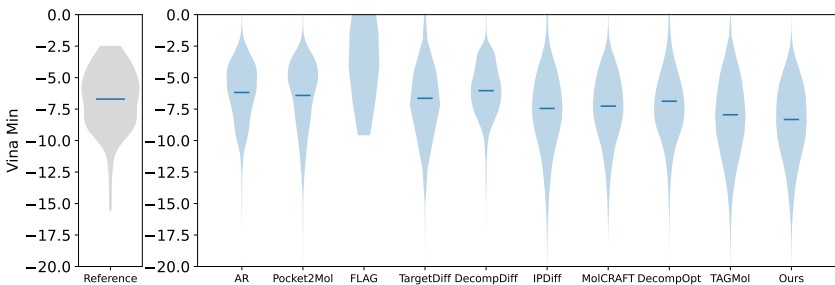

Figure 16: Violin plot for Vina Min distributions of generated molecules and reference molecules.

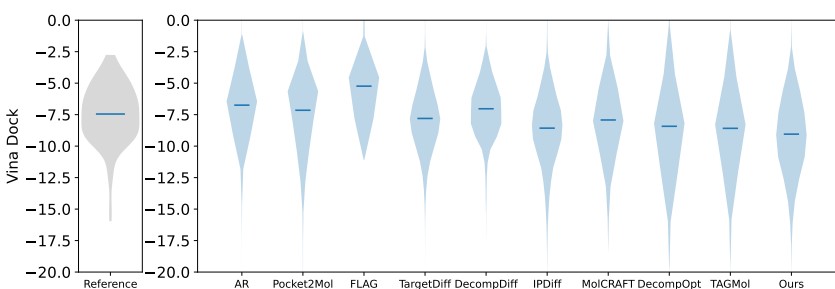

Figure 17: Violin plot for Vina Dock distributions of generated molecules and reference molecules.

Table 11: Proportion (%) of different ring sizes in reference and generated ring structured molecules, where 3-Ring denotes three-membered rings and the like.

|  | #Rings Avg. | 3-Ring | 4-Ring | 5-Ring | 6-Ring |
|---|---|---|---|---|---|
| Reference | 2.8 | 4.0 | 0.0 | 49.0 | 84.0 |
| Train | 3.0 | 3.8 | 0.6 | 56.1 | 90.9 |
| AR | 3.2 | 50.8 | 0.8 | 35.8 | 71.9 |
| Pocket2Mol | 3.0 | 0.3 | 0.1 | 38.0 | 88.6 |
| FLAG | 2.1 | 3.1 | 0.0 | 39.9 | 84.7 |
| TargetDiff | 3.1 | 0.0 | 7.3 | 57.0 | 76.1 |
| DecompDiff | 3.4 | 9.0 | 11.4 | 64.0 | 83.3 |
| IPDiff | 3.4 | 0.0 | 6.4 | 51.0 | 83.7 |
| MolCRAFT | 3.0 | 0.0 | 0.6 | 47.0 | 85.1 |
| DecompOpt | 3.7 | 6.8 | 11.8 | 61.4 | 89.8 |
| TAGMol | 4.0 | 0.0 | 8.5 | 62.5 | 82.6 |
| MolJO (Aff) | 3.6 | 0.0 | 0.4 | 46.7 | 92.5 |
| MolJO (QED) | 3.7 | 0.0 | 0.5 | 58.0 | 96.1 |
| MolJO (SA) | 3.8 | 0.0 | 0.2 | 37.0 | 97.8 |
| MolJO (Aff+SA) | 3.9 | 0.0 | 0.1 | 37.0 | 98.1 |
| MolJO (All) | 3.6 | 0.0 | 0.3 | 44.4 | 97.6 |

Table 12: Inference time cost of optimization baselines.

| Model | Ours | TAGMol | DecompOpt | RGA | AutoGrow4 |
|---|---|---|---|---|---|
| Time (s) | $146 \pm 11$ | $667 \pm 69$ | $11714 \pm 1115$ | $458 \pm 43$ | $2586 \pm 360$ |

common molecule-specific properties such as LogP and QED (Olivecrona et al., 2017; Jin et al., 2018; Nigam et al., 2020; Spiegel & Durrant, 2020; Xie et al., 2021; Bengio et al., 2021), but could be extended to structure-based drug design (SBDD) given docking oracles. However, since most early attempts did not take protein structures into consideration thus were essentially not target-aware, it means that they need to be separately trained on the fly for each protein target when applied to pocket-specific scenarios. RGA (Fu et al., 2022) explicitly models the protein pocket in the design process, overcoming the transferability problem of previous methods.

## J    LIMITATION

Limitation of this work lies in that MolJO adopts no more than three objectives (Affinity, QED, SA) in the optimization process, while there are more objectives that are biologically meaningful such as ADMET. Future directions include extending the general framework to accommodate a wider range of objectives, and even beyond the scope of structure-based molecule optimization (SBMO), for MolJO is a general gradient-based optimization method for continuous and discrete variables, and can be tailored to a great many read-world applications.

