# OpenReview forum: "Unlocking the Power of Gradient Guidance for Structure-Based Molecule Optimization"
_ICLR.cc/2025/Conference — Submitted to ICLR 2025_

### Official Review · Reviewer_F2z6 · 2024-10-29

**Soundness:** 1
**Presentation:** 2
**Contribution:** 1
**Rating:** 5
**Confidence:** 4

**Summary:**

The paper tackles the problem of structure-based molecule optimization (SBMO), i.e. a problem of generating candidate 3D molecules conditioned on a target protein. In contrast to structure-based drug design (SBDD), the generated molecule is also optimized for certain properties. The authors follow related work and model this problem using Bayesian Flow Networks (BFNs), which are designed to capture mixed modalities (discrete atom types and continuous 3D coordinates). The novelty introduced by the authors are (1) gradient guidance, which allows for explicit optimization of properties of interest, and (2) backward correction strategy, which corrects past estimates based on a current optimized one. The authors perform multiple experiments showing superiority of their method compared to various baselines on a variety of different tasks.

**Strengths:**

* I think that the application of gradient guidance to SBMO is a good idea;
* Comprehensive list of baselines;
* Comprehensive list of evaluation tasks.

**Weaknesses:**

1. **Unclear contributions.** The authors claim two main contributions:  (1) Gradient guidance in BFNs with mixed modality applied to SBMO and (2) novel backward correction strategy. Given some of the related work, I do not consider the contributions large enough for a full ICLR paper. Specifically:
  * BFNs have already been applied to 3D molecule modeling [2] and even to structure-based drug design [3, 4] (Note that authors do not cite [4], which is very strongly related to this work and definitely should be discussed).
  * It has been shown that BFNs can be seen as SDEs [1] and therefore it is well-known how to apply guidance there (both classifier-based and classifier-free).
  * The novel backward correction strategy seems to me to be a slight modification of the one suggested in [3].
  * Even the generative "backbone" model is taken from [3] as is without any modifications.
2. I would rather characterize this work as "finetuning" of [3]. Specifically, my assessment of the contributions is:
  * the application of guidance to a pre-trained model (from [3])
  * a slight modification of the variance reduction technique, where instead of the full past, a sliding window is taken (again from [3])
3. **Presentation needs improving.**
  * Some sentences are incomprehensible to me, such as
    * Line 182 "Though different from guided diffusions that operate on noisy latent y, this guidance aligns with our generative process conditioned on θ". What does it mean that guidance aligns with the generative process?
  * The introduction of guidance is the central component of the paper. However, we learn that the method "requires training energy functions as a proxy" in the last paragraph of the paper! I do not understand why the paper is not mostly discussing the energy proxies and how are they defined/trained/evaluated etc. Furthermore, simply adding an energy proxy to any of the baselines would surely improve their performance. Even a modification as simple as: generating multiple candidates and choosing the best one using the energy proxy.
  * I do not understand what Figure 2 is supposed to convey. There is no "take-home" summarizing message. In the text (lines 314-316) we read "it succeeds in balancing sample quality (explore) and optimization efficiency (exploit)". I do not see that in Figure 2. Why is sample quality called "explore"? How is Figure 2 supporting the claim about the tradeoff? The colored lines to me seem random and without any clear pattern.
4. **Lack of mathematical rigor.**
  * Line 187 definition of $\pi$. Is it a purely heuristics based definition? Regular guidance is derived from the Bayes' rule applied to the conditional log density (conditioned on some property of interest). What about this formulation? This is just a multiplication of two densities without an elaboration.
  * Proposition 4.1. I don't understand which parts of the proposition are assumptions and which are the claims. What does it mean that "originally" $\mu_i$ follows a Gaussian?
5. **A very strong objection I have is to the experimental design.** In my opinion it is impossible to assess the quality of the work without more information:
  * What are model sizes (the model proposed by the authors including all its components: the main model, energy proxies, and anything else that needs to be trained; and compare with model sizes of the baselines)
  * What is the training time? (Your method vs baselines)
  * What is the sampling time? (Your method vs baselines)
  * You include your method with a beam search. Perhaps other generative methods would perform even better when equipped with the beam search sampling strategy?
  * Optimization-based baselines. I strongly encourage the authors to include AutoGrow4 [5] as optimization-based baseline. It has been recently reported to work significantly better than RGA [6] (different version of the Vina software was used in that study - has a different range of Vina Dock values)
6. **Reproducibility is questionable.** The code is submitted, but there are no trained model checkpoints provided, so I cannot check the parameter count myself, nor check sampling time or verify the reported results.

---
References

[1] Xue et al. "Unifying Bayesian Flow Networks and Diffusion Models through Stochastic Differential Equations" (ICML2024) - It has been shown that BFNs are equivalent to DMs so deriving guidance for BFNs is not a novel contribuion

[2] Tao et al. "A Bayesian Flow Network Framework for Chemistry Tasks" (Arxiv)

[3] Qu et al. "MolCRAFT: Structure-Based Drug Design in Continuous Parameter Space", (ICML2024)

[4] Song et al. "UNIFIED GENERATIVE MODELING OF 3D MOLECULES VIA BAYESIAN FLOW NETWORKS" (ICLR 2024)

[5] Spiegel et al. "Autogrow4: an open-source genetic algorithm for de novo drug design and lead optimization" (ChemInf 2020)

[6] Karczewski et al. "WHAT AILS GENERATIVE STRUCTURE-BASED DRUG DESIGN: TOO LITTLE OR TOO MUCH EXPRESSIVITY?" (Arxiv)

**Questions:**

* In Line 420 we read "Note that for fair comparison, we restrict the size of generated molecules by reference molecules so that both generative models and optimization methods navigate the similar chemical space," - this in particular means that you use the ground-truth information about the size of the reference molecule? I genuinely appreciate the authors reporting this explicitly. However, this brings a very important question: is this applied to all methods? Were the numbers for all baselines generated by the authors (as opposed to taking them from the publications) with this modification?
* line 052 I do not see this as a weakness of DecompOpt. Some tasks require specific tools.
* Line 077 There's no citation around "gradient guidance" What do authors mean? Classifier-free guidance? Classifier-based guidance? Something else? Based on the later reference, I assume the authors mean classifier-free guidance. However, based on the later parts of the paper I think it is something else.
* Line 87 What are the suboptimal results? E.g. MoFlow [1] adopts this approach and achieves very good empirical performance
* Line 102 what issue of inconsistencies?
* Line 138 just because something is unusual does not mean it cannot work well in practice. See [1] again.
* Line 141 - "often a problematic assumption" Citations? I am aware of works that use this assumption and work well in practice, i.e. [2] again.
* Line 366 - Is the purpose of Figure 3 to show that the introduction of guidance results in a distribution shift? I think that's to be expected. To me, that's rather a sanity check than a strong insight. It would perhaps be more interesting to see how this plot compares with optimization-based methods?
* Figure 5: Why not Vina Dock? I think this one is the most important—also no error bars or statistical significance tests.
* Table 4: What happened to Vina Dock?
* I would like to know more about the "energy proxies":
    * How large are these models - each energy proxy is as large as the TargetDiff model? This seems to make the baseline comparison unfair.
    * What are their sampling times?
    * How accurate are they?
    * How are they trained? Appendix C I think hints at it, but it seems incomplete (equation 21). How is $\theta$ defined for the training purposes? I assume that the protein-ligand complexes are sampled according to the $p_{\text{data}}$ distribution, but how are they then transformed to obtain $\theta$?
* Line 522 typo: "gradiant" -> "gradient"

---
[1] Zang et al "MoFlow: an invertible flow model for generating molecular graphs" (KDD 2020)

---

> ### Author Response · Authors · 2024-11-20
>
> We thank reviewer F2z6 for providing such detailed feedback that helps us improve our manuscript, and we are happy to discuss it further and hopefully clarify some possible misunderstanding.
>
> **Q1: Missing citations**
>
> Thanks for pointing this out, and we've added the missing citations.
>
> **Q2: Equivalence of BFN and Diffusion**
>
> We'd like to point out that BFNs are not equivalent to DMs. Although BFN-Solver [1] tried to mathematically unify BFN and diffusion via SDE, its conclusion is based on the reduction of graphical models (from Fig. 2a to 2b in GeoBFN [4]), i.e., marginalizing out the $y$ in BFN. By doing so, uncertainty, also arguably the essence of Bayesian inference, is not fully reflected in such a reduction. For example Eq. (9) in [1] does not correctly reflect the original loss for continuous data in Eq. (101) in [2], the latter with a different loss weighting of $\frac{1-\gamma(t)}{\gamma(t)}$, which is Var(x) in Eq. (83) and closely related to uncertainty. It's worth noting that [1] completely inverts the BFN and makes it look just like diffusion model by starting from a "forward process" that seemingly comes out of nowhere, which actually is not a prerequisite for BFN in which what needs to be specified is only the prior distribution and Bayesian update rule instead.
>
> A further note is that BFN and diffusion models fundamentally differ in both their training and sampling behaviors. While the original BFN [2] uses a conjugate prior modeled as a Gaussian noisy channel, this does not imply that BFNs are inherently limited to this kind of training. This is one possible instantiation but does not constrain the broader applicability of BFNs. Thus the so-called unified perspective might only be applicable to the overlapping cases with diffusion and BFNs (the Gaussian-distributed parameter), but it is not immediately clear if it still holds when the conjugate prior is something other than Gaussian, for example the von Mises distribution in [3]. As for sampling, the BFN Solver proposed by [1] was found to have issues of diversity in molecule generation [5].
>
> Therefore, we sincerely hope that reviewer F2z6 could adopt a forward-thinking perspective of BFN. BFN represents a generalizable framework rather than a single, narrow implementation, and the foundational elements (e.g. input, output, sender, receiver distributions) do not prescribe any specific implementation. A limited view of its core concept is reductive and might fail to reflect the flexibility inherent in the framework's design.
>
> **Q3: Unclear contributions**
>
> Our contributions lie in both the methodology (deriving gradient guidance within BFNs and proposing a generalized sampling strategy) and in implementing and evaluating SBMO applications. As for reviewer's concerns over the former, we would like to clarify by highlighting the significance of both the guidance within BFN (below) and the backward correction strategy (in Q4).
> Given the answer to Q2, the best form of exerting gradient guidance remains open within the novel framework of BFN. For example, the guidance can be formalized over either latent $y$, the parameter $\theta$, or the sample $x$. While we admit that there is a connection between our choice of guidance over Gaussian distributed variables and diffusions as suggested by Remark 4.2, what such guidance amounts to from the fresh view of BFN remains unexplored. We make such analysis available and show how the gradient redistributes the probability through guided Bayesian update, offering a unique perspective in understanding BFNs, and we believe contextualizing the guidance within BFN is certainly a non-trivial contribution. Furthermore, simply reducing BFN to diffusions and claiming trivialness overlooks the distinct advantages of BFN and discourages deeper exploration of this new generative model class.
>
> **Q4: Novelty of backward correction strategy**
>
> Rather than a slight modification, the backward correction strategy effectively bridges the vanilla Bayesian update from [2] and the variance reduction strategy in [6] by generalizing the concept of update and offering an interpolation. The paper also analyzes and explains why [6] achieves higher sample quality than [1] and explores this trade-off further. Refer to Q8 for more details.
>
> [1] Xue et al. Unifying Bayesian Flow Networks and Diffusion Models through Stochastic Differential Equations.
>
> [2] Graves et al. Bayesian Flow Networks.
>
> [3] https://openreview.net/forum?id=Lz0XW99tE0
>
> [4] Song et al. Unified Generative Modeling of 3D Molecules with Bayesian Flow Networks.
>
> [5] https://github.com/ML-GSAI/BFN-Solver/issues/2
>
> [6] Qu et al. MolCRAFT: Structure-Based Drug Design in Continuous Parameter Space.

---

> > ### Author Response · Authors · 2024-11-20
> >
> > **Q5: No modification of generative backbones**
> >
> > The use of an existing generative backbone from MolCRAFT aligns with our goal: to steer pre-trained models through guidance, rather than develop a new generative model. If reviewer F2z6 intended a specific improvement or modification, we would appreciate further clarification.
> >
> > **Q6: Questions regarding presentation**
> >
> > When we state that "guidance aligns with our generative process conditioned on θ", we mean that the guidance mechanism used in our model works in coordination with the aggregated state θ to drive the generative process towards the desired outputs. Unlike traditional guided diffusions that operate directly on noisy latent variables y, our approach integrates latent guidance into θ, which are later fed into the neural network to estimate the clean data, ensuring that each generative step consistently reflects and optimizes target properties while maintaining alignment with the underlying latent structure. This alignment ensures smooth transformation through generative steps.
> >
> > **Q7: More discussions on energy proxy**
> >
> > We appreciate your interest in the energy proxies used in our approach. The time-dependent energy functions are intended to predict target properties such as binding affinity conditioned on the noisy latent space. The choice of energy proxies matters for model performance, as they provide the guiding signal during the generative process. However, we did not put the emphasis on the special designs of energy proxies, as our core method centers on how to apply gradient guidance given a differentiable oracle.
> >
> > While adding an energy proxy to any baseline may seem straightforward, the value lies in how the energy function is integrated into the BFN framework and how it interacts with Bayesian update steps. Simply equipping generation with filtering would not capture the nuanced property optimization performed during the joint inference process.
> >
> > **Q8: What Figure 2 conveys**
> >
> > Thank you for pointing out the need for clarification. The term "exploration" refers to the phenomenon observed in MolCRAFT's variance reduction strategy (Full B.C. with k=200), where rapidly changing gradients are observed during the generative process to explore a broader range of possibilities, ultimately leading to better sample quality compared to methods that maintain aligned gradients across timesteps (Vanilla B.C. with k=1).
> >
> > By referring to a "trade-off," we mean the balance between:
> > - Exploration: Achieving higher sample quality through rapidly changing gradients that explore the optimal direction.
> > - Exploitation: Leveraging aligned gradients to efficiently use the information gained from previous steps, thereby focusing on refining and optimizing current states.
> >
> > **Q9: Lack of mathematical rigor**
> >
> > We apologize for any confusion caused. For line 187, In the optimization setting, the definition $\pi$ follows the formulation of the product of experts [1, 2]. We have added relevant citations to clarify this in the revised manuscript.
> >
> > For proposition 4.1, we have revised the proposition as requested. The corresponding variable from the original distribution modeled by $p_\phi(\theta)$  is denoted as $\theta$, with the guided version as $\theta^*$. The assumption is that if the original parameters follow Gaussian distributions (instantiated in Eq. 14 and 15), the guided versions can be sampled via Eq. 5 and 6.
> >
> > [1] Hinton et al. Training Products of Experts by Minimizing Contrastive Divergence.
> >
> > [2] Kong et al. Diffusion Models as Constrained Samplers for Optimization with Unknown Constraints.
> >
> > **Q10: More experimental results (model sizes, training and sampling time)**
> >
> > We provide the number of model parameters in the table below, with a comparison with the generative optimization baselines.
> >
> > |  | Backbone | Energy proxy |
> > |---|---|---|
> > | Ours | 2.8M | 2.8M |
> > | TAGMol | 2.8M | 2.9M |
> > | DecompOpt | 5.0M | - |
> >
> > For training time, our MolJO trains 50K steps (batch size = 8) on a 3090, while TAGMol trains 180K steps (batch size = 16) for each of its energy proxies on an A100, and DecompOpt trains 237k steps (batch size = 16) for the entire generative backbone on an A100.
> >
> > For sampling time, we report the time cost for optimization baselines in the table below, which is calculated as the time for sampling a batch of 5 molecules on a single NVIDIA RTX 3090 GPU, averaged over 10 randomly selected test proteins.
> >
> > | Model | Ours | TAGMol | DecompOpt | RGA | AutoGrow4 |
> > |---|---|---|---|---|---|
> > | Time (s) | 146 $\pm$ 11 | 667 $\pm$ 69 | 11714 $\pm$ 1115 | 458 $\pm$ 43 | 2586 $\pm$ 360 |

---

> ### Author Response · Authors · 2024-11-20
>
> **Q11: Questions regarding beam search**
>
> Our method does not employ a default beam search strategy (refer to Line 13 in Table 1). We compare it against both generative baselines and gradient-based methods without this strategy. For baselines such as DecompOpt and RGA, which often generate more samples and select the top-K results, we perform a similar "best-of-N" operation (shown in Line 14 of Table 1) for direct comparison. This ensures a fair evaluation across methods.
>
> **Q12: More optimization baselines**
>
> Thank you for suggesting the inclusion of AutoGrow4 as a powerful baseline. We have included the result in the revised manuscript for a more comprehensive comparison.
>
> **Q13: Questionable reproducibility**
>
> We understand the importance of reproducibility and are committed to ensuring transparency. While the submitted code includes scripts for training classifiers, we have also uploaded trained model checkpoints in the supplementary materials to further support verification.
>
> **Q14: Information about the size of the reference molecule during sampling**
>
> We appreciate your careful review and this important question. To clarify, we do not use ground-truth information about the size of the reference molecule during sampling. Instead, we follow the procedures used in TargetDiff and MolCRAFT, sampling molecule sizes according to a prior distribution derived from the pocket size vs. ligand size in the training data.
>
> **Q15: Weakness of DecompOpt**
>
> Thank you for the feedback. Our intent was to emphasize the potential advantage that atom-level gradient guidance offers over molecule-level oracle functions in certain scenarios. Additionally, DecompOpt's inference time is the slowest among all methods compared (see our response to Q10), which makes it less suitable for large-scale molecular datasets where computational efficiency is critical.
>
> **Q16: Meaning of gradient guidance**
>
> Our formulation of gradient guidance is based on the product of experts (Q9), applicable to both classifier-based and classifier-free guidance. In classifier-free scenarios, the energy function can be viewed as the negative log likelihood estimated by a conditional model for given properties. In practice, our approach resembles classifier guidance since we train property predictors to serve as the energy function. However, since there are no explicit "labels" as conditions in our optimization setting for controllable generation, referring to the energy function as a "classifier" may not be fully appropriate.
>
> **Q17: Challenge of applying Gaussian noise to discrete data**
>
> Our claims in line 87 primarily refer to existing diffusion-based guidance methods for molecular graph generation that operate with continuous Gaussian transition kernels or discrete diffusion over categorical features. These methods can yield suboptimal results due to two key challenges:
> 1. Continuous Representations often rely on dequantization methods that may be less effective in generating discrete features, as noted by DiGress [1] and [2]. This is why they are sometimes perceived as "unnatural" (line 138).
> 2. Discrete Representations, though better capturing the inherent discreteness of data, propagating continuous gradients to discrete variables is not inherently valid [3], which can limit their effectiveness.
>
> In contrast, our framework bridges these gaps by leveraging a structured parameter space derived through Bayesian inference. This enables the continuous gradient guidance of discrete variables, combining the strengths of both continuous and discrete optimization approaches for joint, synchronized optimization.
>
> Furthermore, structure-based molecule optimization (SBMO) requires such joint optimization of continuous and discrete variables. This need is illustrated by the suboptimal performance of existing approaches, such as TAGMol, in Figure 1. Our proposed MolJO framework offers a unified solution that handles both modalities efficiently.
>
> [1] Vignac et al. Digress: Discrete denoising diffusion for graph generation.
>
> [2] Kong et al. Autoregressive Diffusion Model for Graph Generation.
>
> [3] Murata et al. G2D2: Gradient-guided Discrete Diffusion for image inverse problem solving.
>
> **Q18: Inconsistency issue**
>
> Inconsistency might arise across different modalities (continuous-discrete) and different timesteps (within the generative process). For the former, it is empirically demonstrated by the performance boost, and for the latter it is shown in Figure 2 (see also our response to Q8).

---

> ### Author Response · Authors · 2024-11-20
>
> **Q19: Distribution shift**
>
> Yes, Figure 3 primarily shows the general effect of guidance. The purpose here is to show that the guidance mechanism not only modifies the distribution but also surpasses the reference distribution within a reasonable range, thereby addressing and overcoming limitations in data distribution quality to a certain extent. For a more detailed comparison, please refer to Figure 11-17 in Appendix E.1.
>
> **Q20: Vina Dock results**
>
> We appreciate the emphasis on Vina Dock results. While we agree that they provide valuable insights, we chose Vina Score and Vina Min as indicators of binding affinity in ablation studies, consistent with prior SBDD conventions. These metrics offer meaningful approximations while being significantly faster to compute.
>
> Below are the Vina Dock results together with Success Rate:
>
> |  | Methods | Vina Dock Avg. ($\downarrow$) | Vina Dock Med. ($\downarrow$) | QED ($\uparrow$) | SA ($\uparrow$) | Success Rate ($\uparrow$) |
> |:---:|:---:|:---:|:---:|:---:|:---:|:---:|
> | w/o Guide | Vanilla (k=1) | -7.37 | -7.31 | 0.46 | 0.62 | 12.2\% |
> |  | Vanilla MC (k=1) | -8.07 | -8.02 | 0.51 | 0.61 | 15.6\% |
> |  | Full B.C. (k=200) | -7.91 | -7.82 | 0.49 | 0.68 | 24.4\% |
> |  | B.C. (k=130) | -7.95 | -7.87 | 0.49 | 0.69 | 25.6\% |
> | w/ Guide | Vanilla (k=1) | -7.49 | -7.46 | 0.46 | 0.62 | 15.1\% |
> |  | Vanilla MC (k=1) | -8.62 | -8.82 | 0.55 | 0.65 | 26.8\% |
> |  | Full B.C. (k=200) | -9.10 | -9.14 | 0.54 | 0.76 | 44.0\% |
> |  | B.C. (k=130) | **-9.11** | **-9.25** | **0.56** | **0.77** | **51.8\%** |
>
> For statistical significance, we report the p-values of pairwise T-test between B.C. (k=130) and Vanilla (k=1), Vanilla MC (k=1), Full B.C. (k=200), respectively. It can be seen that our proposed strategy shows statistically significant improvement w.r.t. Vanilla (p<<0.05), Vanilla MC (p<<0.05 except for QED,  which is not included as an objective in our implementation). Compared with Full B.C., it's true that Vina affinities remain close between the two and shows no significant difference, but the improvement in molecular properties such as QED and SA is clear (p<0.05), showing a better consistency across continuous and discrete modalities and thereby resulting in a higher Success Rate (51.8% vs. 44.0%).
>
> | Metric | vs. Vanilla | vs. Vanilla MC | vs. Full B.C. |
> |---|---|---|---|
> | Vina Score | 2.63e-13 | 4.94e-3 | 7.56e-1 |
> | Vina Min | 3.31e-31 | 3.05e-3 | 9.70e-1 |
> | Vina Dock | 2.79e-35 | 2.91e-3 | 9.07e-1 |
> | SA | 8.10e-115 | 4.66e-75 | 5.56e-3 |
> | QED | 1.98e-26 | 2.06e-1 | 1.14e-2 |
>
> **Q21: More about energy proxies**
>
> In order to make a fair comparison, we stick to the same setting as our gradient-based counterpart TAGMol, where each energy proxy is indeed as large as TargetDiff backbone (2.8M parameters). Additionally, we are actively exploring if reducing the model size of energy proxy has a notable impact on performance.
>
> For sampling of energy proxies, we report the overall time in Q10.
>
> For accuracy of the time-dependent energy proxies, we report the Mean Absolute Error (MAE) in the table below, averaged over 10 evenly divided time intervals.
>
> For training of the energy functions, $\theta$ is defined the same way as backbone MolCRAFT, where the parameter space is defined by $\sigma_1=0.03, \beta_1=1.5$, as described in Appendix C, with the transformation in Eq. 4 in MolCRAFT.
>
> |  | Affinity | QED | SA |
> |---|---|---|---|
> | MAE | 0.086 | 0.017 | 0.007 |
>
> **Q22: Typos in paper**
>
> We have corrected the typo, and we'd like to thank reviewer F2z6 again for the time and efforts devoted to reviewing our paper. Sincerely, we welcome any further questions and hope the reviewer is willing to reassess our paper based on the above answers.

---

> ### Author Response · Authors · 2024-11-25
> **Gentle Reminder**
>
> Dear Reviewer F2z6,
>
> We appreciate your insightful feedback, which has been instrumental in improving our manuscript.
>
> Please let us know if we have adequately addressed your concerns. Your further comments are most welcome. Thank you again for your invaluable contribution to our research.
>
> Warm Regards,
>
> The Authors

---

> > ### Comment · Reviewer_F2z6 · 2024-11-25
> >
> > I thank the authors for their response. I think some of my concerns have been addressed, but not all. I elaborate below.
> >
> > * *By doing so, uncertainty, also arguably the essence of Bayesian inference, is not fully reflected in such a reduction*
> >
> > Is uncertainty modeling crucial for the method proposed by the authors? I have not seen this in the manuscript. I understand that the formalism is different and the Bayesian approach may additionally offer uncertainty estimates, but it does not seem to me like this is relevant in MolJo. Therefore, my concern remains. My understanding is that (perhaps at the cost of lack of uncertainty modeling), BFNs can be interpreted as SDEs and thus we know how to guide them. Have the authors explored this direction? If the authors believe that their approach is better, then it should be demonstrated at least empirically.
> >
> > * *the BFN Solver proposed by [1] was found to have issues of diversity*
> >
> > I do not consider a comment on github to be strong evidence. All the above suggests to me that the natural next step for the authors is to prove the claims above by doing a proper evaluation, i.e. inclusion of the SDE approach in the comparison and also, the "regular guidance" (when we treat the BFN as an SDE).
> >
> > * *simply reducing BFN to diffusions and claiming trivialness overlooks the distinct advantages of BFN*
> >
> > I agree that this would be a great argument if this was demonstrated in the manuscript, but I do not see this evidence. Authors suggest that the Bayesian approach and consequently, uncertainty estimates are what makes BFNs superior to diffusion. But these are not at all discussed in the manuscript, nor used in the proposed method. If the authors also did guidance in a "regular" way by treating the BFN as diffusion and empirically showed massive improvements just by introducing the Bayesian treatment then the gains would be clear.
> >
> > * *works in coordination* and *each generative step consistently reflects and optimizes target properties while maintaining alignment with the underlying latent structure*
> >
> > I am sorry, but these sentences are a bit vague to me. I don't immediately understand what they are supposed to convey.
> >
> > * *However, we did not put the emphasis on the special designs of energy proxies, as our core method centers on how to apply gradient guidance given a differentiable oracle.*
> >
> > On the one hand, yes, but the paper is being positioned as achieving ground-breaking results in the SBMO domain, and therefore the implementation matters in my opinion.
> >
> > * *Exploration: Achieving higher sample quality through rapidly changing gradients that explore the optimal direction.*
> >
> > Exploration explores the optimal direction? I thought that exploitation would be focused on following the optimal direction. This explanation did not make this clearer for me.
> >
> > * *MolJO trains 50K steps*
> >
> > I think GPU time would also be helpful. Do the authors include time to train MolCRAFT? Since other methods need to train from scratch.
> >
> > * *Questions regarding beam search*
> >
> > I feel like it is an odd comparison to include MolJo (N=10) in the comparison. Unless the authors want to include other Generative models and include their performance when equipped with an oracle? This goes back to my concern raised in the review.
> >
> > * *AutoGrow4*
> >
> > I thank the authors for including AutoGrow4. I have some questions though. Did it use the same oracle as MolJo? I find it surprising that MolJo performs so much better in terms of QED than AutoGrow. Especially, since it has been reported, e.g. in [1] that Vina and QED are negatively correlated. So it kind of looks to me like AutoGrow may have been optimized for Vina, while MolJo for the combination of the two. What was the computational budget of AutoGrow? How many generations? By the way, why does DecompDiff have quite a bit worse performance in the table provided by the authors compared to the original publication? I did not carefully check all the baselines.
> >
> > * *Checkpoints*
> >
> > Thank you for uploading checkpoints.
> >
> > * *Size of molecules*
> >
> > Thank you for the clarification. It's important to make it clear in the paper.
> >
> > * *These methods can yield suboptimal results*
> >
> > That is a reasonable claim. However, as I mentioned in the review, there are works that use this assumption and perform well.
> >
> > ---
> >
> > [1] Karczewski et al. "WHAT AILS GENERATIVE STRUCTURE-BASED DRUG DESIGN: TOO LITTLE OR TOO MUCH EXPRESSIVITY?" (Arxiv)

---

> > > ### Author Response · Authors · 2024-11-27
> > > **Response to follow-up questions (1/2)**
> > >
> > > **Q1: Regular SDE guidance**
> > >
> > > Thanks for this insightful suggestion, and we agree that a comparison with the regular guidance is necessary for empirically showing the improvements of our proposed guidance within the BFN framework over simply treating it as SDE.
> > > We implemented the regular guidance (denoted SDE w/ RG below) and report the results in the table below. The objectives we've chosen are the same with our main experiments (i.e. Affinity + SA).
> > >
> > > It can be seen that tuning the guidance scale only optimizes the affinities yet leaves the QED, SA unchanged, suggesting that directing viewing BFN as SDE struggles to optimize the discrete types, for in Table 8, Appendix F.2 it is observed that those molecular properties have more to do with the gradient over type.
> > >
> > > These results indicate that while SDE-based regular guidance can enhance affinity (Vina scores), it struggles to optimize discrete molecular types (QED, SA). This suggests that the SDE formulation might not effectively support a smooth and continuous information flow from the gradient to the aggregated belief z, especially after applying nonlinear transformations. Our approach, specifically designed for BFNs, demonstrates consistent improvements across all molecular properties.
> > >
> > > |  | scale | Vina Score Avg | Vina Score Med | Vina Min Avg | Vina Min Med | QED | SA | Connected |
> > > |---|---|---|---|---|---|---|---|---|
> > > | SDE w/o guide | 0 | -6.59 | -7.07 | -7.30 | -7.21 | 0.49 | 0.64 | 97.8% |
> > > | SDE w/ RG | 0.1 | -6.90 | -6.98 | -7.32 | -7.16 | 0.50 | 0.65 | 97.4% |
> > > |  | 1 | -7.08 | -7.40 | -7.50 | -7.50 | 0.48 | 0.64 | 96.2% |
> > > |  | 5 | -7.63 | -8.02 | -8.06 | -8.04 | 0.49 | 0.63 | 77.8% |
> > > |  | 10 | -6.91 | -8.51 | -8.39 | -8.68 | 0.44 | 0.58 | 43.0% |
> > > | Ours | 50 | -7.52 | -8.02 | -8.33 | -8.34 | 0.56 | 0.78 | 97.0% |
> > > |  | L2-norm | -7.37 | -7.64 | -8.09 | -7.95 | 0.56 | 0.76 | 99.0% |
> > >
> > > **Q2: Explanation for guidance that "aligns with our generative process conditioned on θ"**
> > >
> > > We apologize for the confusion. By guidance to "align with our generative process conditioned on θ", we mean that θ is fed into the neural network to inform its prediction of generated sample, i.e. θ as the parameters of its input distribution $p_I(x | \theta)$, and the gradient for guidance is obtained from the energy function defined over θ so as to guide it. We believe that such guidance over θ can effectively inform the generative process.
> > >
> > > **Q3: Special designs of energy proxies**
> > >
> > > Thanks for acknowledging MolJO's performance in SBMO tasks. We will add a more detailed discussion of the definition / training / evaluation of energy proxies in our revised manuscript.
> > >
> > > **Q4: Exploration that explores the optimal direction**
> > >
> > > Thank you for your question. You’re correct that exploitation involves following the current local optimal direction to refine solutions, i.e. following a greedy strategy. Sorry for the inaccurate description of "explores the optimal direction", and what we actually meant by "optimal" is more of "globally optimal". As far as we know, exploration refers to the process of searching more broadly across the solution space to avoid premature convergence to local optima.
> > > - In our method, rapidly changing gradients facilitate exploration by allowing the model to traverse different regions of the solution space, identifying potentially better directions or solutions.
> > > - Conversely, aligned gradients reflect exploitation, where the model leverages previously acquired information to converge more efficiently towards a refined solution.
> > >
> > > This balance between exploration (diversity) and exploitation (refinement) ensures that the model not only explores a wide range of possibilities but also effectively optimizes the best candidates it discovers.
> > >
> > > **Q5: Training time**
> > >
> > > Each energy function is trained to converge within 9 hours, yet a direct comparison in training time would not be immediately possible, given that different sized GPUs are used (our 40GB 3090 vs. others' 80GB A100) and batch sizes also differ.
> > >
> > > Regarding the training of backbone, TAGMol directly adopts the pretrained TargetDiff, and we've built on top of the pretrained MolCRAFT, while DecompOpt needs to train a new generative model from scratch.
> > >
> > > **Q6: Questions regarding beam search**
> > >
> > > Thanks for your suggestions, and that is a reasonable concern. We're currently actively working on it, and will update the results as soon as they are available.

---

> > > > ### Author Response · Authors · 2024-11-27
> > > > **Response to follow-up questions (2/2)**
> > > >
> > > > **Q7: Baseline performances**
> > > >
> > > > Regarding AutoGrow4, we follow its default configurations in its official GitHub without further modifications. While other works might have observed a negative correlation between QED and Vina in their methods, our proposed MolJO succeeds in optimizing Vina without compromising QED (see Table 8), and we did not include QED as one of the objectives, and chose Vina + SA instead.
> > > >
> > > > Regarding DecompDiff, Line 7 in Table 1 is reported for its variant in setting its prior mode to ref, which means it directly utilizes the ground truth size of reference molecules, while Table 2 reports its opt-prior version that aligns with its original paper.
> > > >
> > > > We'd like to emphasize that all the evaluations conducted here faithfully adhere to the official implementation or officially released samples for reproduction, and we're happy to clarify any doubts.
> > > >
> > > > **Q8: Size of molecules**
> > > >
> > > > Thanks for the suggestion, and we will surely add this to our draft.
> > > >
> > > > **Q9: Works that use this assumption and perform well**
> > > >
> > > > Thanks for sharing this important work with us. SimpleSBDD is quite novel in that it decomposes the generation of 3D geometry (an unlabeled graph) and corresponding atom types in labeling it. This is methodologically different than simultaneously generating both continuous coordinates in 3D space and discrete atom types. While we agree that SimpleSBDD reflects a promising direction, again we'd like to e mphasize that this does not harm our main claim that it is challenging to simultaneously optimize continuous-discrete variables, and that we have figured out one possible solution.
> > > >
> > > > Overall, we'd like to thank reviewer F2z6 again for the detailed discussion that motivated us to better reflect on our method, and we felt lucky to have such thoughtful reviews. We welcome any further comments or suggestions.

---

> > > > > ### Comment · Reviewer_F2z6 · 2024-11-28
> > > > >
> > > > > I thank the authors for the continued and constructive discussion and I appreciate the effort to conduct additional analyses. I have a few more questions.
> > > > >
> > > > > 1. Comparison with regular SDE
> > > > >
> > > > > Could you please give more details on how exactly you set this baseline up? My thinking was: Perform regular SDE modelling in $\theta$ (latent) space, which is how I understand the unification of the BFN and SDE framework as developed by [1]. Can the authors confirm that is what has been done? Either way, it would be helpful for me if you could provide more details. My thinking was that if the discrete component was encoded in the latent space the model could also learn to optimize it, but it looks like it's even worse than no guidance, which is surprising to me.
> > > > >
> > > > > Furthermore, the table the authors provided is not that clear to me. Where is the boundary between the SDE and BFN? Is it one row for "SDE w/o guide", 4 rows for "SDE w/ RG" and two rows for "Ours"?
> > > > >
> > > > > 2. Re beam search
> > > > >
> > > > > Thank you, I am curious to see the results.
> > > > >
> > > > > ---
> > > > >
> > > > > [1] Xue et al. Unifying Bayesian Flow Networks and Diffusion Models through Stochastic Differential Equations.

---

> ### Author Response · Authors · 2024-11-28
>
> We are happy to further discuss with reviewer F2z6 and update the results.
>
> **Q1: Regular SDE**
>
> Yes, we followed [1] and borrowed the code for SDE Euler update for both continuous and discrete data. The sampling is done by discretizing the following form of SDE:
> $$\mathrm{d}{\bf{x}} = [F(t){\bf{x}} - G(t)^2 \nabla_{{\bf{x}}}\log p_t({\bf{x}})] \mathrm{d}t + G(t)\mathrm{d}\bar{{\bf{w}}}$$
> where the $\bf{x}$ corresponds to $\bf{\mu}$ for continuous data, and $\bf{y}$ for discrete data. The estimated score function is given as
> $$s({\bf{\mu}}(t), t)=-\frac{1}{\sqrt{\gamma(t)(1-\gamma(t))}}\epsilon({\bf{\mu}}(t), t), \quad s({\bf{y}}(t), t)=-\frac{{\bf{y}}(t)}{K\beta(t)}+\epsilon_s({\bf{y}}(t), t)-\frac{1}{K}$$
>
> A few changes have been made since our backbone MolCRAFT is not trained to predict $\epsilon$, but $x_0$ instead, thus we recalculated the coefficient for updating continuous $\mu$ by $\hat{x}$ and empirically verified the modification.
>
> For classifier guidance, initially we simply subtract the estimated score function by the scaled gradient $s\nabla_{\bf{x}}\log p_E({\bf{x}})$, and we apologize for possible miscalculation of the scale of gradient. We have updated the results in the table below, with a clearer comparison between SDE-based and BFN-based sampling. It shows that SDE-based guidance requires 5x sampling time (i.e. the same discrete-time steps as in training) while being slightly inferior in optimizing both objectives (Affinity + SA).
>
> |  | Scale | Sample Steps | Vina Score Avg | Vina Score Med | Vina Min Avg | Vina Min Med | QED | SA | Connected |
> |---|---|---|---|---|---|---|---|---|---|
> | SDE w/o guide | 0 | 1000 | -6.59 | -7.07 | -7.30 | -7.21 | 0.49 | 0.64 | 97.8% |
> | BFN w/o guide (B.C. k=130) | 0 | 200 | -6.50 | -7.00 | -7.03 | -7.14 | 0.49 | 0.69 | 97.4% |
> | SDE w/ RG | 50 | 1000 | -7.16 | -7.51 | -7.75 | -7.73 | 0.55 | 0.71 | 97.8% |
> | BFN w/ guide (B.C. k=130) | 50 | 200 | -7.52 | -8.02 | -8.33 | -8.34 | 0.56 | 0.78 | 97.0% |
>
> **Q2: Beam search results for generative models**
>
> Thanks for your interest. We have made the same top-of-N selection based on z-score for picking the top 1/10 out of roughly 100 generated molecules w.r.t. each of the 100 test proteins (a tenth of 10,000 in all), and provided the result in the table below. This generally shows what the "concentrated space" for desirable drug-like candidates looks like for generative models. Supplementing it with Table 1, we found that the performance of DecompOpt is comparable to a top-of-10 version of DecompDiff, and ours to that of MolCRAFT in terms of success rate. Considering that the sampling time of DecompDiff (\~309s) and MolCRAFT (\~22s), DecompOpt (\~11714s) is significantly heavier than ours (\~146s), and does not offer additional benefits in computational efficiency compared to a top-of-10 DecompDiff.
>
> | Method | Vina Score Avg | Vina Score Med | Vina Min Avg | Vina Min Med | Vina Dock Avg | Vina Dock Med | QED | SA | Diversity | Success Rate |
> |---|---|---|---|---|---|---|---|---|---|---|
> | AR | -6.71 | -6.35 | -7.12 | -6.63 | -7.81 | -7.33 | 0.64 | 0.70 | 0.60 | 19.07% |
> | Pocket2Mol | -5.80 | -5.39 | -7.18 | -6.50 | -8.32 | -7.78 | 0.67 | 0.84 | 0.59 | 40.54% |
> | FLAG | 50.37 | 43.14 | 6.27 | -3.38 | -6.57 | -6.47 | 0.74 | 0.78 | 0.71 | 9.60% |
> | TargetDiff | -7.06 | -7.57 | -8.10 | -8.11 | -9.31 | -9.18 | 0.64 | 0.65 | 0.67 | 32.60% |
> | DecompDiff | -5.78 | -5.82 | -6.73 | -6.57 | -8.07 | -8.03 | 0.61 | 0.74 | 0.61 | 32.10% |
> | MolCRAFT | -7.54 | -7.89 | -8.40 | -8.13 | -9.36 | -9.05 | 0.65 | 0.77 | 0.63 | 55.00% |
> | IPDiff | -8.15 | -8.67 | -9.36 | -9.27 | -10.65 | -10.17 | 0.60 | 0.62 | 0.69 | 34.60% |

---

> > ### Comment · Reviewer_F2z6 · 2024-11-29
> >
> > I thank the authors for the additional clarifications and results. I believe these should make it into the document's final version and be discussed properly. Perhaps, the authors might even want to investigate a bit further what exactly makes BFNs superior to latent diffusion models in the SBMO context. The empirical evidence suggests that the superiority of BFNs over regular SDEs is not that large (especially given the large variance of the Vina Scores in general; and that the diffusion model was not optimally tuned to solve this task, i.e. hyperparameters etc.), which makes the narrative of BFNs being ideally suited for SMBO not as strong in my opinion.
> >
> > That said, I think this experiment strengthens the paper, albeit with some rewriting (I do not believe that simply putting the SDE experiment in the Appendix is a good choice because it should be properly discussed in the main text). I will raise my score to 5.

---

> > > ### Author Response · Authors · 2024-11-29
> > >
> > > We thank reviewer F2z6 for these actionable insights, and we will take them into careful consideration and keep on investigating the superiority of BFNs in terms of the reduced sampling time, the nonlinear transformation $f$ for discrete data and its impact on gradient guidance, etc. We are actively working on more analysis experiments and a more thorough revision.

---

### Official Review · Reviewer_LDEf · 2024-10-30

**Soundness:** 2
**Presentation:** 2
**Contribution:** 2
**Rating:** 5
**Confidence:** 2

**Summary:**

The paper deals with generative models for molecules. Specifically, it proposes a way to improve generation quality by using gradient guidance, based on a specified energy function, for both continuous and discrete variables (atoms’ positions and types, respectively). The paper builds on Bayesian Flow Networks, which operates on some continuous latent variables, which facilitates guidance across both data modalities.

**Strengths:**

The authors tackle the problem of applying gradient guidance jointly for discrete and continuous variables in the context of molecule generation. This is a challenging task, as naively applying gradient guidance to discrete variables is not possible. I think this problem is quite relevant, as proper use of different guidance techniques has been observed to lead to improved performances across many domains.

To the best of my knowledge, the method proposed by the authors, which relies on Bayesian Flow Networks and applying guidance in some underlying continuous variables, is novel. And empirical results appear to be strong.

**Weaknesses:**

While I understand the core idea and problem in the paper, I find the details hard to follow, including exactly what variables represent and how are the update rules obtained. Please see the “questions” section below for extended details.

**Questions:**

I’m having difficulties following the method’s explanation, what variables represent, and how updates and equations are derived. A few examples below:

- [BFN in preliminaries.] “The receiver holds a prior belief $\theta_0$, and updates…” Exactly what does $\theta$ represent? Is it pointwise parameters, is there a distribution defined over them?

- [BFN in preliminaries.] Eq (1) shows a distribution over $\theta_i$, but eq. (2) shows $\theta$ as a deterministic variable (given observations y_0, …, y_i). I understand Eq (1) is the posterior given a datapoint $x$, integrating out potential observations, but Eq (2) represents the deterministic updates we get for some given observations at different noise levels?

- The parameters $\theta$ are fed through a NN to model output distribution over clean data. Why is it sensible to apply guidance over theta (if they are connected to the clean samples through a NN, and the energy is typically defined over clean samples)? Coming back to the question above, what do these latents represent?

- [Proposition 4.1.] Is $\mu_\phi$ an output of the NN $\Phi(\theta_{i-1})$? Also, in the definition of $\theta$ (line 201) $\theta=[\mu, z]$ you use $z$, but Eq (6) uses $y$, stating “recall $z=f(y)$”. I suppose this comes from Eq (2)? Are $y$ the noisy observations? If so, why is Eq (6) sampling $y$? (Should it be sampling $z$ which is part of $\theta$?)

- [Proposition 4.1.] Where are the original Gaussians from line 207 coming from (in-line equations, just before Eq (5))? Why are those the correct “unguided kernels” $p_\phi$?

- After reading section 4.1, it is unclear to me how samples are actually generated by the model without using backward correction.

- [Line 233.] This line uses the notation $e_v$ without definition. $e_v$ is defined later in line 240. I’d suggest to introduce variables before using them in equations. That same paragraph states “Surprising as it may seem, this is mathematically grounded…” in which way?

I know some of these questions are not related to the core method but are more general. But I think it would be good for the paper to be self-contained. Asking for a fully fledged description of BFNs in the main paper may be unrealistic, but introducing the necessary components, even briefly, that lead to the equations being used later on, would be good. Unfortunately, I cannot recommend acceptance for the paper in its current form. I’m open to revisiting my score if the paper is updated addressing these general comments (or if they are clarified during the discussion, if I’m missing/misunderstanding something).

A few additional questions.

- [Prop 4.1.] “it suffices to sample guided … [guided Gaussians]”. These expressions are based on a 1st order Taylor expansion. Would these become increasingly exact as the updates become smaller? (Related to the discretization used?) I think the approximation used here could be briefly discussed in the main text, as it is claimed before, in line 192, that the analytic expressions for the guided kernels are derived.

- [Detail in Line 187.] The definition of $\pi(\theta_i | \theta_{i-1})$ should have a $\propto$ instead of $=$? If $p_\phi$ and $p_E$ are both normalized, their product is not necessarily normalized. For instance, consider the product of two Gaussian densities, the resulting thing is not normalized.

---

> ### Author Response · Authors · 2024-11-20
>
> Thanks for the thorough review, and especially for the general advice of introducing the necessary components  of BFN in more detail. We have added a more intuitive explanation in Appendix A, and addressed the remaining questions below. Please feel free to raise any further questions, which will definitely help us improve our manuscript.
>
> **Q1: BFN notations**
>
> > [BFN in preliminaries.] “The receiver holds a prior belief $\theta_0$, and updates…” Exactly what does $\theta$ represent? Are these pointwise parameters? Is there a distribution defined over them?
>
> Yes, that's correct. We denote $\mathbf{\theta}=[\theta^{(1)},…,\theta^{(N)}]$ as the parameters of an N-dimensional factorized distribution, where the input distribution is defined by $p(m|\theta) = \prod_{d=1}^N p(m^{(d)} | \theta^{(d)})$. This formulation allows $\theta$ to encapsulate intrinsic parameters across different dimensions and modalities.
>
> Specifically, these parameters can be pointwise vectors in $\mathbb{R}^3$ for N continuous atom coordinates (representing positions in 3D space) and on the probability simplex in $\mathbb{R}^K$ for N discrete atom types (defining categorical distributions over K possible types).
>
> Together, these parameters define a factorized distribution over N atoms. This structure enables joint optimization and parameterization of molecular structures across different modalities within the Bayesian Flow Network framework.
>
> > [BFN in preliminaries.] Eq (1) shows a distribution over $\theta_i$, but eq. (2) shows $\theta$ as a deterministic variable (given observations y_0, …, y_i). I understand Eq (1) is the posterior given a datapoint $x$, integrating out potential observations, but Eq (2) represents the deterministic updates we get for some given observations at different noise levels?
>
> Thanks for pointing this out. It's true that Eq. 1 represents the posterior distribution over $\theta_i$, given a data point $x$. In this formulation, we integrate out potential observations $y$ over the sender distribution $p_S$ to model the uncertainty of the latent state.
>
> On the other hand, Eq. 2 represents deterministic updates of $\theta$ when given specific observations of $y$ at different noise levels. This deterministic update reflects how the receiver distribution $p_R$ evolves as it processes known observations, resulting in a more precise latent state estimation for $\theta$ under specific conditions.
>
> During training, since the sender distribution is known, we can integrate out $y$ within $p_S$. However, during sampling, we only work with the receiver distribution $p_R$. This leads to different forms of the function mapping $f$ that governs updates, as discussed in the additional discussion provided in Appendix A.
>
> **Q2: Reason for guidance over $\theta$**
> > The parameters $\theta$ are fed through a NN to model output distribution over clean data. Why is it sensible to apply guidance over theta (if they are connected to the clean samples through a NN, and the energy is typically defined over clean samples)? Coming back to the question above, what do these latents represent?
>
> Our approach applies guidance over $\theta$ because the energy function is defined on the intermediate latents. This enables us to guide the aggregated latent states within the generative trajectory, which are then used to predict clean datapoints, rather than directly working with clean samples. By doing so, we ensure consistent updates that reflect both observed noise and the desired target properties.
>
> The latent $y$ can be viewed as the observed noisy representations of clean data, and $\theta$ a posterior belief updated by noisy observations, therefore can be understood as aggregated $y$. For more intuitive discussion of how it is derived through Bayesian inference, please refer to Appendix A.

---

> ### Author Response · Authors · 2024-11-20
>
> **Q3: Guidance notations**
> > [Proposition 4.1.] Is $\mu_\phi$ an output of the NN $\Phi(\theta_{i-1})$? Also, in the definition of $\theta$ (line 201)  you use $\theta=[\mu, z]$, but Eq (6) uses $y$, stating “recall $z=f(y)$”. I suppose this comes from Eq (2)? Are the $y$ noisy observations? If so, why is Eq (6) sampling $y$? (Should it be sampling $z$ which is part of $\theta$?)
>
> Yes,  $\mu_\phi$ is indeed one of the outputs of the neural network, as it parameterizes the distribution over clean coordinates.
>
> Regarding the use of $\theta=[\mu, z]$ and subsequent references to $y$ in Eq. 6, and $y$ do represent noisy observations drawn during training and guidance stages. The function $z=f(y)$ defines a mapping that links these noisy observations to their discrete state representations. The reason why Eq 6. samples $y$ is actually related with numerical stability: for continuous variables, the noisy samples aggregate linearly into $\mu$ with a constant multiplier, as demonstrated in Remark 4.2. But for discrete variables, the gradient w.r.t. $z$ needs to be multiplied by $[\frac{\partial h}{\partial y}]^{-1}=\frac{1}{z_i(1-z_i)}$ that actually differs along each dimension, and due to the fact that $z$ lies on the probability simplex and will be increasingly sharp towards the end of generation (i.e. Approaching a one-hot), such a gradient would be highly unstable.
>
> **Q4: Gaussian distribution in Proposition**
>
> > [Proposition 4.1.] Where are the original Gaussians from line 207 coming from (in-line equations, just before Eq (5))? Why are those the correct “unguided kernels” $p_\phi$?
>
> Sorry for the confusion caused. Proposition 4.1 actually assumes that the unguided distributions are Gaussian, while they are then fulfilled in Eq. 14, 15. We have revised the proposition to make it more clear.
>
> **Q5: Sample generation**
>
> > After reading section 4.1, it is unclear to me how samples are actually generated by the model without using backward correction.
>
> We apologize for any confusion regarding sample generation. A more detailed overview of the Bayesian Flow Network (BFN), including an intuitive description of its generation process without backward correction, has been provided in Appendix A for clarity.
>
> **Q6: More about sender distribution for discrete data**
>
> > [Line 233.] This line uses the notation $e_v$ without definition. $e_v$ is defined later in line 240. I’d suggest to introduce variables before using them in equations. That same paragraph states “Surprising as it may seem, this is mathematically grounded…” in which way?
>
> Thank you for pointing out the use of the notation $e_v$. We have revised the draft to introduce it earlier, ensuring clarity and consistency. The statement for mathematical ground highlights how the formulation of sender distribution for discrete data is constructed with theoretical rigor, as opposed to heuristics that tried to relax the discrete variable into some continuous space. To illustrate, continuous diffusion applies Gaussian noise to the one-hot encoding, while BFN constructs a "multi-hot" y that follows a multinomial distribution stemming from the categorical. Details can be found in Appendix B.
>
> **Q7: Higher-order approximation of guided distributions**
>
> > [Prop 4.1.] “it suffices to sample guided … [guided Gaussians]”. These expressions are based on a 1st order Taylor expansion. Would these become increasingly exact as the updates become smaller? (Related to the discretization used?) I think the approximation used here could be briefly discussed in the main text, as it is claimed before, in line 192, that the analytic expressions for the guided kernels are derived.
>
> Thanks for the suggestion! We have added the approximation used in line 192, mentioning that the guided form is based on a first-order Taylor expansion, and indeed, higher-order terms would theoretically improve the approximation, yet second-order terms would require additional time complexity such as computing a Hessian matrix, and often lead to instability during sampling, which we will leave for future work. Here we briefly list a possible solution for sampling $(\mu_i, y_i) \sim \mathcal{N} \left(\begin{pmatrix} \mu_i^* \\\ y_i^*\end{pmatrix}, \Sigma_{\pi} \right)$, where the covariance matrix $\Sigma_{\pi} =
> \begin{pmatrix} \sigma + H_{\mu\mu}^{-1} & H_{\mu y}^{-1} \\\ H_{\mu y}^{-1} & \sigma' + H_{yy}^{-1} \end{pmatrix}$, and $H_{\mu\mu}$ ($H_{yy}$) is the Hessian matrix of second derivatives with respect to $\mu$ ($y$), $H_{\mu y}$  is the cross-derivatives.
>
> **Q8: Definition of guided $\pi$**
>
> > [Detail in Line 187.] The definition of $\pi(\theta_i | \theta_{i-1})$ should have a $\propto$ instead of $=$? If $p_\phi$ and $p_E$ are both normalized, their product is not necessarily normalized. For instance, consider the product of two Gaussian densities, the resulting thing is not normalized.
>
> Yes. Thank you for pointing out the typo in the definition. We have corrected this in our revised manuscript.

---

> ### Author Response · Authors · 2024-11-25
> **Gentle Reminder**
>
> Dear Reviewer LDEf,
>
> We appreciate your insightful feedback, which has been instrumental in improving our manuscript.
>
> Please let us know if we have adequately addressed your concerns. Your further comments are most welcome. Thank you again for your invaluable contribution to our research.
>
> Warm Regards,
>
> The Authors

---

> > ### Comment · Reviewer_LDEf · 2024-11-26
> > **Thank you for the response**
> >
> > I'd like to thank the authors for their response and insights. After reading the response, I still choose to keep my score. I still believe clarity can be improved in the paper, including some of the comments above (and comments from other reviewers), a more thorough discussion of the energies, their applications to the parameters theta, etc.
> >
> > For instance, the response above mentions "Our approach applies guidance over because the energy function is defined on the intermediate latents." This could use a more rigorous explanation in the paper; how are these energies obtained (they are trained, as mentioned in the final part of the paper, how? How costly is this?), how are they defined over these intermediate latents theta. While the method appears to yield good empirical performance, in my opinion more extensive changes to the presentation would be needed.

---

### Official Review · Reviewer_t4Np · 2024-11-04

**Soundness:** 2
**Presentation:** 2
**Contribution:** 2
**Rating:** 3
**Confidence:** 4

**Summary:**

This paper considers the gradient guidance of generative models in the structure-based drug design problem. Specifically, this paper proposes a new method that handles both gradients to update the discrete atom token space and the continuous coordinate space. An additional backward correction strategy is proposed to improve the efficiency of the optimization process. Effectiveness of the proposed method is validated over a set of optimization setups including structure-based drug design, multi-objective optimization and substructure-constrained optimization.

**Strengths:**

* Equipping molecular generative models, in particular diffusion and flow models, with conditional generation and optimization capability is important yet under-studied.
* The proposed backward correction method to balance exploration and exploitation is novel.
* Experiments are conducted on a wide range of optimization setups which show the promise of the proposed method.

**Weaknesses:**

* This paper misses a large chunk of literature on molecular optimization, see this review paper [1].
* Some claims in the paper are not appropriate, previous structure-based drug design models also optimize both the atom type and coordinate [2, 3] (gradient-based algorithm and evolutionary algorithm). I am not sure if it is appropriate to call it the *first proof-of-concept for gradient-based optimization of continuous-discrete variables* because (1) the scope is very narrow, (2) other papers have worked on it [2, 3], (3) it is unclear why we have to use a gradient-based method, even for discrete probability distributions, and (4) the "gradient" for the discrete case is not a gradient but a weighting (which can relate to derivative-free optimization/sampling method such as sequential Monte Carlo).
* Some notations are unclear.

[1] Du, Y., Jamasb, A.R., Guo, J., Fu, T., Harris, C., Wang, Y., Duan, C., Liò, P., Schwaller, P. and Blundell, T.L., 2024. Machine learning-aided generative molecular design. Nature Machine Intelligence, pp.1-16.

[2] Lee, S., Jo, J. and Hwang, S.J., 2023, July. Exploring chemical space with score-based out-of-distribution generation. In International Conference on Machine Learning (pp. 18872-18892). PMLR.

[3] Schneuing, A., Harris, C., Du, Y., Didi, K., Jamasb, A., Igashov, I., Du, W., Gomes, C., Blundell, T., Lio, P. and Welling, M., 2022. Structure-based drug design with equivariant diffusion models. arXiv preprint arXiv:2210.13695.

**Questions:**

* What is the parameter $\phi$ and how do you learn it?
* How is the time-dependent energy function learned? Since you only know the property when $t=0$.
* What are $\mu$ and $y$ in proposition 4.1? $\theta$ was first defined as $[\mu, z]$ but later defined as $[\mu, z = f(y)]$ while $f$ is not defined.
* In eq. 8, it is a little confusing gradient over $y^*$ vs proposition 4.1 gradient of $y$.
* In eq. 8, how is the chain rule performed? What is the dependence of E over $\mu$ vs $h$?
* What is $\sigma'$?
* One more suggestion is to have a broader discussion of related work about conditional generation in diffusion/flow models.
* The backward correction idea is interesting, does it also connect to the resampling trick or restart sampling? [1, 2]
* In eq. 12, by the linearity of Gaussian, what's the difference between predicting $\hat{x}$ from $\theta_{i-1}$ vs $\theta_{i-2}$?
* I am happy to raise my score if the authors clarify some of my concerns.

[1] Lugmayr, A., Danelljan, M., Romero, A., Yu, F., Timofte, R. and Van Gool, L., 2022. Repaint: Inpainting using denoising diffusion probabilistic models. In Proceedings of the IEEE/CVF conference on computer vision and pattern recognition (pp. 11461-11471).

[2] Xu, Y., Deng, M., Cheng, X., Tian, Y., Liu, Z. and Jaakkola, T., 2023. Restart sampling for improving generative processes. Advances in Neural Information Processing Systems, 36, pp.76806-76838.

---

> ### Author Response · Authors · 2024-11-20
>
> Thanks for the detailed feedback! While we appreciate the reviewer’s recognition of the importance of equipping molecular generative models with conditional generation and optimization capabilities, we would like to first clarify that MolJO is not based on diffusion or flow models. Instead, our method leverages Bayesian Flow Networks (BFNs), which represent a distinct paradigm for generative modeling, and enable more structured and flexible updates, particularly suited for joint continuous-discrete optimization task. We believe that our approach offers a valuable perspective on this challenge, and we have added a more detailed introduction of BFN in Appendix A.
>
> **W1: Missing literature of molecule optimization**
>
> Thanks for the reference to the review paper, which helps us better understand the literature of molecule optimization. We have added the related works in Appendix I.
>
> **W2.1: Clarification of the scope**
>
> > Some claims in the paper are not appropriate, previous structure-based drug design models also optimize both the atom type and coordinate [2, 3] (gradient-based algorithm and evolutionary algorithm). I am not sure if it is appropriate to call it the first proof-of-concept for gradient-based optimization of continuous-discrete variables because (1) the scope is very narrow, (2) other papers have worked on it [2, 3]
>
> Thanks for raising this important question.
> We acknowledge that previous models have also made significant strides and addressed the optimization of both atom types and coordinates. However, our method introduces a unique approach within the novel generative framework of BFN through synchronized, joint gradient optimization within a unified differentiable space, which distinguishes it from prior models, such as those described in references [2] and [3], in tackling the challenge of optimizing both atom types and coordinates.
> - Evolutionary-based DiffSBDD [3] relies on iterative search for optimized properties, selecting top-K molecules to seed the new population. It uses supervision signals only implicitly, and we aim to open the door to directly utilizing the atom-level gradient signals.
> - Gradient-based algorithm [2] builds on GDSS [4] for graph generation but does not incorporate SE(3)-equivariance in its modeling. This approach treats the graph features as continuous variables, which means that it dequantizes node vectors and adjacency matrices using uniform or Gaussian noise and subsequently learns Gaussian diffusion models over these dequantized variables. While such dequantization and Gaussian diffusions can capture some aspects of graph structures, they face inherent challenges with compatibility between continuous Gaussian diffusion processes and discrete data, making it difficult for the generation of consistent and valid graph structures [6].
>
> Moreover, when it comes to optimizing both coordinates and types, more sophisticated designs are required to ensure the consistency across different modalities, such as those nuanced noise schedules and multiple stages in MolDiff [5], thus it can be expected that potential suboptimality could arise from indirect guidance approaches.
>
> Unlike prior works that typically treat discrete variables as continuous or optimize the entire molecule via resampling, our work uniquely emphasizes explicit, synchronized, joint gradient-guided optimization across continuous and discrete variables with SE(3)-equivariance. This represents a fresh perspective compared to existing methods, and ensures smooth and consistent information flow across modalities.
>
> (For better readability, the index for reference [2, 3] remains consistent with the original question.)
>
> [2] Lee et al. Exploring chemical space with score-based out-of-distribution generation.
>
> [3] Schneuing et al. Structure-based drug design with equivariant diffusion models.
>
> [4] Jo et al. Score-based Generative Modeling of Graphs via the System of Stochastic Differential Equations.
>
> [5] Peng et al. MolDiff: Addressing the Atom-Bond Inconsistency Problem in 3D Molecule Diffusion Generation.
>
> [6] Vignac et al. DiGress: Discrete Denoising diffusion models for graph generation.

---

> > ### Author Response · Authors · 2024-11-20
> >
> > **W2.2: Gradients for discrete data**
> >
> > > (3) it is unclear why we have to use a gradient-based method, even for discrete probability distributions, and (4) the "gradient" for the discrete case is not a gradient but a weighting (which can relate to derivative-free optimization/sampling method such as sequential Monte Carlo).
> >
> > For (3), the reason for a gradient-based method centers on better controllability and efficiency compared to gradient-free sampling methods, which often require initiating and maintaining multiple sampling chains. By providing dense, atom-level gradient signals, our approach allows for more precise and controllable generation, enhancing the structural accuracy and optimization capabilities, and is well-suited for structure-based molecule optimization (SBMO).
> >
> > However, prior methods rely on heuristic assumptions to approximate gradients for discrete data, for example assumptions that view the graph G with $n$ nodes and $n \times n$ matrix as a continuous tensor of order $n+n^2$ so that the gradient w.r.t. G can be defined [6].
> > We take this further by naturally integrating continuous information flow into the optimization of discrete variables. By associating discrete variables with continuous parameters, we ensure gradients are meaningful in a differentiable parameter space without the need to assume or approximate the entire graph as a continuous tensor or so.
> >
> > For (4), we'd like to clarify the interpretation of gradient for discrete data. While gradients for discrete variables might appear as weighting functions, this interpretation does not diminish their nature as gradients acting on parameters within the optimization process, in a way that maintains mathematical and practical alignment with gradient-based guidance.

---

> > > ### Author Response · Authors · 2024-11-20
> > >
> > > **W3: Unclear notations**
> > >
> > > We apologize for the confusion caused, and we shall address any ambiguity in the following answers.
> > >
> > > > What is the parameter $\phi$ and how do you learn it?
> > >
> > > $\phi$ is associated with the generative model's backbone and represents the trainable parameters of the neural network. Within the BFN framework, the neural network operates over the aggregated latent representation $\theta$ (factorized along each dimension) to model the interdependence and predict the clean data from noisy observations.
> > >
> > > The parameters $\phi$ are learned by optimizing a loss function that minimizes the KL divergence between the sender and receiver distributions during the generative process (Eq. 4). Note that it is done in the generative backbone MolCRAFT, and we plug in a differentiable energy function to steer the generated distribution towards desirable directions.
> > >
> > > > How is the time-dependent energy function learned? Since you only know the property when t=0.
> > >
> > > The energy function $E(\theta, t)$ is part of a Boltzmann distribution over parameters $\theta$. Although the property is known at t=0, the Bayesian inference defines the space of parameters $\theta$for any given timestep $i$. Thus, we associate the property value to every $\theta_i$ throughout different timesteps, allowing time-dependent properties to be predicted during intermediate stages of the generative optimization process.
> > >
> > > > What are $\mu$ and $y$ in proposition 4.1? $\theta$ was first defined as $[\mu, z]$ but later defined as $[\mu, z=f(y)]$  while $f$ is not defined.
> > >
> > > In Proposition 4.1, $y$ represents latent variables ($y^x$ and $y^v$ for continuous and discrete modalities, respectively, here we omit the $\cdot^v$ for brevity), and $\theta=[\mu, z]$ are the aggregated beliefs over these two modalities of the molecule structure, where $\mu$ corresponds to continuous one. We refer to $z$as a functional mapping applied to the noisy latent $y$, explicitly showing the link between $y$ (the noisy representation) and the aggregated $z$ (the underlying belief over actual probability mass function for discrete data) as shown in Eq. 2.
> > >
> > > > In eq. 8, it is a little confusing gradient over $y^*$ vs proposition 4.1 gradient of $y$.
> > >
> > > Thanks for pointing this out. We actually made a slight abuse of notation in dropping the superscript $\cdot^v$ for the noisy latent of discrete data (in Proposition 4.1), while keeping those to separate the latent for continuous (Eq. 8) and discrete variables (Eq. 9, 10). We are actively conducting a revision of our notations in our manuscript.
> > >
> > > > In eq. 8, how is the chain rule performed? What is the dependence of E over $\mu$ vs $h$?
> > >
> > > The chain rule in Eq. 8 relates the gradient over noisy latent $y$ to the gradient over $\mu$ (belief of parameters for continuous coordinates) through the Bayesian update function $h$ (Eq. 7).
> > >
> > > > What is $\sigma'$?
> > >
> > > Sorry for the confusion, here $\sigma, \sigma'$denotes a variance term for the isotropic Gaussian distribution of the continuous and discrete variables in Proposition 4.1, respectively. And the reason why it is isotropic is that the parameters for input distribution (Appendix A) are independent from other dimensions by definition, resulting in a factorized distribution. Specifically, $\sigma'$ corresponds to the variance term $\Delta\beta' K \mathbf{I}$ in Eq.15 for the distribution of $y$.
> > >
> > > **Q1: Broader discussion of related work**
> > >
> > > > One more suggestion is to have a broader discussion of related work about conditional generation in diffusion/flow models.
> > >
> > > We appreciate the suggestion to discuss related conditional generation work in diffusion/flow models more broadly. Unlike standard diffusion models that often rely on a forward noising processes, or flow models that utilize invertible transformations, BFNs are grounded in a non-autoregressive framework where generative modeling is conceptualized as a message exchange between a sender and receiver (details in Appendix A). This enables more structured and flexible Bayesian updates in a unified parameter space, particularly suited for joint continuous-discrete optimization tasks. The strength of our approach lies in incorporating gradient guidance within a BFN-based generative space, where conditional generation and optimization are achieved through continuous Bayesian updates and a backward correction strategy. This differs fundamentally from conditional generation approaches in diffusion or flow models, which typically operate on different assumptions, priors, and update mechanisms.

---

> > > > ### Author Response · Authors · 2024-11-20
> > > >
> > > > **Q2: More on backward correction**
> > > >
> > > > > The backward correction idea is interesting, does it also connect to the resampling trick or restart sampling? [1, 2]
> > > >
> > > > Thanks for recognizing the novelty of backward correction strategy.
> > > >
> > > > Unlike the resampling trick or restart sampling that involves repeatedly adding noise to achieve semantic consistency, our backward correction does not involve any additional steps that would prolong the generative process.
> > > >
> > > > Instead, "backward correction" means redefining the "previous belief" of parameters  $\theta_i$ from $\theta_{i-1}$  to an earlier state $\theta_{i-k}$. This essentially creates a new starting point for the chain without requiring extra steps, which is directly linked to the Bayesian aggregation of the posterior belief, as illustrated in Figure 1D of the paper. The "backward" aspect refers to altering the starting belief state within the Bayesian update process, enhancing the accuracy of latent variable estimates without redundant iterations.
> > > >
> > > > > In eq. 12, by the linearity of Gaussian, what's the difference between predicting $\hat{x}$ from $\theta_{i-1}$ vs $\theta_{i-2}$?$
> > > >
> > > > Thanks for the detailed observation, and it's true that by the linearity of Gaussian, we can obtain the same distribution of $\theta_{i-1}$ from any intermediate $\theta_{0:i-2}$ through Bayesian update, (i.e., $p_U(\theta_{i-1} | \theta_{i-2}, x)=p_U(\theta_{i-1} | \theta_k, x), \forall k=0, \dots, i-3$).
> > > >
> > > > However, there is a crucial distinction when predicting  $\hat{x}$ from $\theta$ at different levels of accuracy. Since $\theta_{i-1}$  reflects a more recent state with lower uncertainty than $\theta_{i-2}$, predictions from $\theta_{i-1}$  ($\hat{x}_{i-1}$) are expected to be more accurate with the optimized properties. Moreover, $\theta_{i-1}$ is further optimized towards the desired target properties, reinforcing its strength in estimating clean datapoint.
> > > >
> > > > Essentially, Eq. 12 emphasizes that the optimal information in $\theta_{i-1}$ should be leveraged to overwrite previous Bayesian updates, thereby preserving accuracy and optimizing the predictions in a simulation-free manner.

---

> ### Comment · Reviewer_t4Np · 2024-11-23
> **Thanks for the discussion**
>
> Thank you for writing the rebuttal, I have a couple of more comments/questions
>
> First of all, I agree with what you said, but every paper needs to be at least somehow different to be published, but not every paper claims to be the first of its kind because of its difference or novelty. The idea is essentially, by given so many relevant works in this field, I suggest removing the phrase "first proof of concept", I think it makes more sense to be used if very few works have been done in the field or if it provides a highly novel concept which potentially people have not thought it could work or nothing close to it worked before ("proof of concept").
>
> Re the word "gradient", I am not too strongly against the word, as long as you can find a good root in the derivative-free and discrete optimization literature so it doesn't confuse people.
>
> Re time-dependent energy function, the reason why I ask is that to achieve exact guidance, it always requires a time-dependent energy function, in the diffusion model literature, it is usually evaluated by sampling from p(x_0|x_t), \log \sum exp(-energy function). I am trying to understand if you did it this way or you simply obtain it through the same function by evaluate on different x_t and t which is often inexact.

---

> > ### Author Response · Authors · 2024-11-29
> > **Gentle Reminder**
> >
> > Dear Reviewer t4Np,
> >
> > We appreciate your insightful feedback, which has been instrumental in improving our manuscript.
> >
> > Please let us know if we have adequately addressed your concerns. Your further comments are most welcome. Thank you again for your invaluable contribution to our research.
> >
> > Warm Regards,
> >
> > The Authors

---

> ### Author Response · Authors · 2024-11-23
> **Thanks for the feedback**
>
> Thank you for this thoughtful suggestion. We agree that the phrase may imply a broader claim than intended, and our goal was to highlight the unique integration of gradient guidance for joint continuous-discrete optimization within Bayesian Flow Network (BFN). We recognize that emphasizing our contributions in terms of methodological advances and practical implications rather than positioning it as the "first" will better reflect its value, and we have revised the draft accordingly.
>
> Regarding the use of "gradient", we agree that ensuring clarity is crucial, especially when applying gradient-based methods to discrete variables. We will be careful in using "gradient" in this context.
>
> Regarding time-dependent energy function, we trained a predictor $E(\theta_t, t)$ directly over the parameter space (the belief as aggregated noisy samples) to estimate the $\nabla_{\theta_t} p_E(\theta_t)$, given $\theta_t$ at different accuracy levels. Then, we obtain the guidance term from the same function when sampling $\theta_t$, which proves to be effective.
>
> Our guidance operates in a training-based manner, essentially different from the training-free methods that estimate the time-dependent term through a loss function defined over clean data space, which is challenging and usually requires multiple model evaluations for the true posterior mean $\hat{x}_0$ (i.e. sampling $p(x_0|x_t)$ many times) [1] or multiple back propagations with the log-mean-exp-negative operation [2]. One such training-free guided molecular diffusion [3] based on DPS [1] requires turning off the guidance during the early "chaotic stage" and applying the MC sampling and the time-travel technique to improve $\hat{x}_0$ estimates, while our guided Bayesian update provides uncertainty-aware guidance with reduced variance without needing such adjustments.
>
> We believe that our work offers a promising foundation for further exploration of guidance within BFN, and we look forward to more exciting discoveries in this emerging class of generative models.
>
> [1] Diffusion Posterior Sampling for General Noisy Inverse Problems.
>
> [2] Loss-guided diffusion models for plug-and-play controllable generation.
>
> [3] Training-free Multi-objective Diffusion Model for 3D Molecule Generation.

---

### Official Review · Reviewer_C1Hy · 2024-11-10

**Soundness:** 3
**Presentation:** 2
**Contribution:** 2
**Rating:** 6
**Confidence:** 2

**Summary:**

MolJO introduces a framework for structure-based molecule optimization that handles both continuous (atomic coordinates) and discrete (atom types) molecular properties through gradient guidance and Bayesian inference. The method achieves state-of-the-art results on the CrossDocked2020 benchmark with a 51.3% Success Rate and 4× improvement over previous gradient-based methods, while maintaining SE(3)-equivariance. Using a backward correction strategy and joint optimization approach, MolJO demonstrates superior performance across various drug design tasks, though currently limited to three main objectives (Affinity, QED, SA).

**Strengths:**

Technical Innovation: The paper presents an approach to jointly optimize both continuous (atomic coordinates) and discrete (atom types) variables in molecule optimization.

Strong Performance: The method achieves impressive results, showing a 4× improvement over gradient-based baselines and 2× better "Me-Better" ratio than 3D baselines, while maintaining SE(3)-equivariance. The success rate of 51.3% on CrossDocked2020 represents a significant advance.

Practical Application: The method demonstrates strong versatility across real drug design tasks like R-group optimization and scaffold hopping, and effectively balances multiple objectives while generating valid molecular structures. The backward correction strategy also provides a practical way to balance exploration and exploitation during optimization.

**Weaknesses:**

Presentation: The paper is based on BFNs. A self-contained introduction may help the reader understands the proposed method better. A comparison with discrete diffusion and generative flow net would also be helpful.

Limited Objective Scope: The method is only validated on three objectives (Affinity, QED, SA) despite the wide range of important molecular properties in drug discovery. The paper does not explore crucial biological objectives or demonstrate how the approach would scale to more objectives.

Computational Analysis Gaps: The paper lacks detailed analysis of computational requirements and efficiency. There is insufficient discussion about how the backward correction window size affects computational costs, and no clear comparison of computational resources needed versus other methods.

Hyperparameter Sensitivity: The method's performance appears sensitive to key hyperparameters like guidance scale and correction window size, but the paper does not provide clear guidelines for selecting these parameters or analyze their impact systematically. This raises questions about the method's robustness in practical applications.

**Questions:**

How reliable is the oracle for real-world drug discovery? Is the cost of calling the oracle a concern?

How does the backward correction window size impact performance versus computational cost, and what determines the optimal balance?

Can this joint optimization approach extend beyond the three basic properties (Affinity, QED, SA) to handle more complex molecular properties relevant to drug discovery?

---

> ### Author Response · Authors · 2024-11-20
>
> We appreciate the constructive feedback from reviewer C1Hy as well as the efforts in reviewing our paper. We hope that the concerns have been addressed below, and we welcome any further questions.
>
> **W1: Presentation**
>
> Thanks for the advice! We've added more explanations of BFN in Appendix A. Briefly speaking, the major difference between BFN and discrete diffusion / GFlowNets lies in that the latter two operate in a discrete state space instead of a continuous space of noisy observations $y$. For BFN over discrete data, there is no need to utilize a transition matrix (as in discrete diffusion) or a (state, action) pair (as in GFlowNets) to navigate the probability simplex, enjoying the benefit of smoother transformation.
>
> **W2: Limited Objective Scope**
>
> We admit that other biological objectives are yet to be tested within MolJO framework, but an extension would not be difficult. We've added the Lipinski's Rule of Five (RO5) objective. Please see Q3 below. If there are specific objectives that reviewer C1Hy would like to see, we will definitely give it a try in our revision.
>
> As for scaling to $k$ objectives, it requires $k$ gradients that either come from neural networks to make the predictions, or some other calculations, so in the worst case the time scales linearly to the number of objectives. However, due to the fact that different objectives are usually not equally hard and so will the network parameters and calculation time, the actual time cost depends on the implementation.
>
> **W3: Computational Analysis Gaps**
>
> We've added the comparison of computation time for optimization baselines in Appendix H, which is calculated as the time for sampling a batch of 5 molecules on a single NVIDIA RTX 3090 GPU, averaged over 10 randomly selected test proteins.
>
> | Model | Ours | TAGMol | DecompOpt | RGA |
> |---|---|---|---|---|
> | Time (s) | 146 $\pm$ 11 | 667 $\pm$ 69 | 11714 $\pm$ 1115 | 458 $\pm$ 43 |
>
> **W4: Hyperparameter Sensitivity**
>
> Thanks for bringing this up, which inspires us to investigate key hyperparameters like guidance scale and correction window size further.
>
> For guidance scale, we found that the gradient norm is generally between 0.02 and 0.06, which might explain why scale = 50 appears optimal in the previous study. We conduct a new experiment and adopt the L2 gradient normalization that automatically adjusts the guidance scale and thus does not require manual tuning. The updated result is shown in the table below, and it can be seen that our guidance method remains robust to the normalized gradient scale.
>
> For correction window size, it can be seen from Figure 9, Appendix F.2 that the performance of backward correction strategy remains robust when the window size k > 50, i.e., correcting a sufficient part (more than 1/4) of the optimization history. This brings consistent performance boost.
>
> |  | Vina Score Avg. | Vina Score Med. | Vina Min Avg. | Vina Min Med. | QED | SA | Connected |
> |---|---|---|---|---|---|---|---|
> | Reference | -6.36 | -6.46 | -6.71 | -6.49 | 0.48 | 0.73 | 100% |
> | w/o guide | -6.40 | -6.89 | -7.33 | -7.23 | 0.48 | 0.67 | 95.8% |
> | w/ guide (s=50) | -7.52 | -8.02 | -8.33 | -8.34 | 0.56 | 0.78 | 97.0% |
> | w/ guide (grad norm) | -7.37 | -7.64 | -8.09 | -7.95 | 0.56 | 0.76 | 99.0% |
>
> **Q1: How reliable is the oracle for real-world drug discovery? Is the cost of calling the oracle a concern?**
>
> This is a great question. The reliability of using oracles for real-world drug discovery, particularly for Vina scores as affinity, remains a topic of discussion. The most reliable method is definitely wet-lab experiments, yet it is often infeasible in the early-stages. While AutoDock Vina may not be perfect, it is one of the most widely used open-source docking tool, offering a balance between speed and accuracy for in-silico calculation [1]. Thus, the oracle is of considerable value for benchmarking molecular properties for drug discovery.
>
> As for the cost, Vina Dock usually takes tens of seconds per molecule (exhaustiveness=16), and therefore, in order to call the oracles and evaluate 10K molecules in the main experiment, it requires more than 10 hours to run on a single CPU. This becomes increasingly costly when the optimization methods like RGA typically run several rounds before arriving at the final outputs, making it a CPU-heavy process.
>
> [1] Su et al. Comparative Assessment of Scoring Functions: The CASF-2016 Update.

---

> ### Author Response · Authors · 2024-11-20
>
> **Q2: How does the backward correction window size impact performance versus computational cost, and what determines the optimal balance?**
>
> Thank you for raising this question, which helps us in clarifying our proposed method further. The size of backward correction window actually does not have a notable impact on computational cost, because we've derived the simulation-free solution for the backward corrected Bayesian update (Eq. 14, 15), thus the update does not require traversing the window of size k $(i-k, i-k+1, \dots, i-1)$, but instead jumps from $i-k$ toward the next step $I$.
>
> **Q3: Can this joint optimization approach extend beyond the three basic properties (Affinity, QED, SA) to handle more complex molecular properties relevant to drug discovery?**
>
> Yes, certainly.  As shown in the table below, we conduct an experiment that solely optimizes Lipinski's Rule of Five (RO5), and our guidance method succeeds in optimizing molecular properties such as QED, SA without compromising Vina Affinities, even though they are not listed as optimization objectives.
>
> We admit that for more complex molecular properties, such as ADMET and pharmacokinetics, richer datasets that contain such information will be needed in order to build a good data-driven differentiable proxy.
>
> |           | Vina Score Avg. | Vina Score Med. | Vina Min Avg. | Vina Min Med. | QED  | SA   | Connected |
> |-----------|-----------------|-----------------|---------------|---------------|------|------|-----------|
> | Reference | -6.36           | -6.46           | -6.71         | -6.49         | 0.48 | 0.73 | 100%      |
> | w/o guide | -6.40           | -6.89           | -7.33         | -7.23         | 0.48 | 0.67 | 95.8%     |
> | w/ RO5       | -6.30           | -7.06           | -7.14         | -7.37         | 0.58 | 0.70 | 98.0%     |

---

> > ### Comment · Reviewer_C1Hy · 2024-11-25
> >
> > Thank authors for the detailed reply, which has addressed my questions and comments. I keep my positive rating.

---

### Meta-Review · Area_Chair_d2GC · 2024-12-26

**Metareview:**

This paper extends Bayesian flow networks, and propose a gradient guidance based on an energy function for both continuous and discrete variables in the context of molecular generation for the position and type of atoms. The proposed method achieves state-of-the-art results on the CrossDocked2020 benchmark with a 51.3% Success Rate and 4× improvement over previous gradient-based methods, while maintaining SE(3)-equivariance.  The objectives validating these claims are on affinity QED and SA.  The paper was discussed at length between authors and the reviewers, who provided great feedback.

While the paper improved over the rebuttal, several issues remain unresolved and would need a major revision of the paper to address and a new round of reviews, please see below:
* Lack of clarity, many of the notations and the key element of the methods needs clarification and a primer on BFN at the beginning of the paper would be helpful
* A more thorough explanations of the energies defined, their applications on the latent representation the computational cost
* Clarifying the scope of the paper and how to position it with respect to the literature addressing similar problems as raised by reviewer t4Np
* Incorporating new results on regular SDE and beam search to the main paper, along success rates and docking results as suggested by reviewer F2z6
We encourage the authors to take into account reviewers feedback and resubmit the paper to the next venue.

**Additional Comments On Reviewer Discussion:**

Please see above where we summarize the discussion and the rebuttal of this paper.

---

### Decision · Program_Chairs · 2025-01-22

Reject